# Combatting Dimensional Collapse in LLM Pre-Training Data via Diversified File Selection

**Ziqing Fan[1,2,\*], Siyuan Du[2,3,\*], Shengchao Hu[1,2], Pingjie Wang[1,2],**
**Li Shen[4,✉], Ya Zhang[1,2],Dacheng Tao[5], Yanfeng Wang[1,2,✉]**
[1]Shanghai Jiao Tong University, China; [2]Shanghai AI Laboratory, China; [3]Fudan University, China
[4]Shenzhen Campus of Sun Yat-sen University, China; [5]Nanyang Technological University, Singapore
`zqfan_knight@sjtu.edu.cn, dusiyuan@pjlab.org.cn, charles-hu@sjtu.edu.cn,`
`pingjiewang@sjtu.edu.cn, mathshenli@gmail.com, ya_zhang@sjtu.edu.cn,`
`dacheng.tao@ntu.edu.sg, wangyanfeng@sjtu.edu.cn`

## Abstract

Selecting high-quality pre-training data for large language models (LLMs) is crucial for enhancing their overall performance under limited computation budget, improving both training and sample efficiency. Recent advancements in file selection primarily rely on using an existing or trained proxy model to assess the similarity of samples to a target domain, such as high quality sources BookCorpus and Wikipedia. However, upon revisiting these methods, the domain-similarity selection criteria demonstrates a diversity dilemma, i.e. *dimensional collapse* in the feature space, improving performance on the domain-related tasks but causing severe degradation on generic performance. To prevent collapse and enhance diversity, we propose a **Di**ver**S**ified **F**ile selection algorithm (**DiSF**), which selects the most decorrelated text files in the feature space. We approach this with a classical greedy algorithm to achieve more uniform eigenvalues in the feature covariance matrix of the selected texts, analyzing its approximation to the optimal solution under a formulation of $\gamma$-weakly submodular optimization problem. Empirically, we establish a benchmark and conduct extensive experiments on the TinyLlama architecture with models from 120M to 1.1B parameters. Evaluating across nine tasks from the Harness framework, DiSF demonstrates a significant improvement on overall performance. Specifically, DiSF *saves 98.5% of 590M training files* in SlimPajama, outperforming the full-data pre-training[1] within a 50B training budget, and achieving about *1.5x training efficiency* and *5x data efficiency*. Source code is available at: https://github.com/MediaBrain-SJTU/DiSF.

## 1 Introduction

Pre-trained Large Language Models (LLMs) have demonstrated remarkable capabilities (Brown, 2020a; Chowdhery et al., 2023a; Touvron et al., 2023b), but their training is computationally expensive, with costs increasing as model size and training data grow (Rae et al., 2021; Patterson et al., 2021; Thoppilan et al., 2022). For instance, training GPT-3 with 175 billion parameters is estimated to produce 552 tons of $CO_2$ emissions and consume 1,287 MWh of energy (Patterson et al., 2021). In practice of commercial use and common academic research, training budgets such as the number of pre-trained tokens are typically predefined, determined by available devices and training time constraints (Hoffmann et al., 2022). To optimize the performance of LLMs within the budget, selecting high-quality pre-training data from large text corpora is essential, boosting both training and sample efficiency.

Recent innovations of selecting files for pre-training LLMs mostly rely on using an existing or trained proxy model and designing a proxy function to access the similarity to a target domain, which is regarded as high-quality data. Heuristic classification (Brown, 2020b; Chowdhery et al., 2023b) trains a binary classifier and select similar content to text domains like Books and Wikipedia (Computer,

---

[1]Full-data pre-training in this paper refers to pre-training the LLMs on all training files in SlimPajama until the specified training budget is reached.



| (a) Heuristic | (b) DSIR | (c) QuRating-W | (d) D4 | (e) DISF |

Figure 1: The t-SNE (Van der Maaten & Hinton, 2008) visualization of text features (normalized to the unit sphere) selected by different methods on SlimPajama. We use Contriever (Izacard et al., 2021) to extract features. (a) and (b) show Heuristic classification and DSIR based on the Wikipedia and Book domains, while (c) depicts QuRating based on writing judgments. We visualize top 500 text features selected by their criterion, which forms a long narrow band, indicating dimensional collapse. (d) and (e) represent D4 and our DiSF. For D4, we display 500 random samples after reducing redundancy, while for DiSF, we select samples with the highest values based on equation 6. Both methods, especially DiSF, show more uniformly scattered features, indicating improved diversity.

2023). DSIR (Xie et al., 2023b) also targets these domains, using a hashed n-gram extractor to measure similarity. QuRating (Wettig et al., 2024) leverages GPT-3.5-turbo to train a judge model that evaluates the quality of domains like writing and education. However, as shown in Figure 2, these methods based on specific domains lead to a diversity dilemma, known as *dimensional collapse* in representation learning (Jing et al., 2022; Zbontar et al., 2021; Bardes et al., 2022; Shi et al., 2022; Fan et al., 2024). Their feature vectors of samples span a lower-dimensional subspace, indicating less diversity, improving performance in domain related tasks, such as reading comprehension (e.g., ARC (Clark et al., 2018) and OBQA (Mihaylov et al., 2018)), but causing severe degradation in overall performance across diverse domains, particularly in physical world knowledge tasks like PIQA (Bisk et al., 2020) and HellaSwag (Zellers et al., 2019).

In our paper, we revisit these algorithms by visualizing the feature representations of their selected text samples as shown in Figure 1, and performing eigenvalue analysis on the features' covariance matrix (see Section 2 for details). As depicted in Figure 1 (a), (b), and (c), we observe dimensional collapse, where text samples selected based on a specific domain show dominant top eigenvalues, indicating long narrow feature spaces. In contrast, text samples selected using our diversified method exhibit less dominant eigenvalues, leading to greater uniformity across feature dimensions (Figure 1 (e)). Recent methods, such as D4 (Tirumala et al., 2023) and INGENIOUS (Renduchintala et al., 1991), recognize the importance of diversity and select informative samples by leveraging feature distances and similarity kernels, respectively. However, they fall short in achieving the level of uniform representations attained by our approach as shown in Figure 1 (d) and (e) and Figure 3.

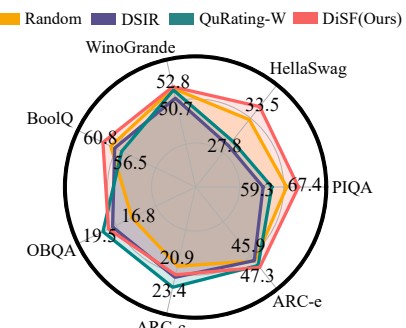

Figure 2: Commonsense reasoning ability of pre-trained TinyLlama 1B using various selection methods evaluated on seven tasks of Harness. DSIR uses Wikipedia and BookCorpus as high quality source and QuRating-W selects samples with writing style score. All methods select 1.5% of training files in SlimPajama and pre-train 50B tokens.

To prevent collapse and enhance diversity, we propose DiSF, that selects files by minimizing the Frobenius norm of the features' covariance matrix, achieving more uniform eigenvalues. We approach this with the classical greedy algorithm and analyze its approximation to the optimal solution under the formulation of $\gamma$-weakly submodular optimization problem (DAS, 2011). Empirically, we establish a benchmark on the newly released and popular TinyLlama (Zhang et al., 2024a) architecture with models of 120M, 560M, and 1.1B parameters. Extensive experiments and ablation studies, conducted across nine tasks on the Harness framework, demonstrate the superior general performance of our method compared to baselines. Specifically, out of 590M training files in SlimPajama (Touvron et al., 2023a; Computer, 2023), our DiSF selects just 1.5% (about 9B training tokens), outperforming full-data pre-training under 50B training budget, and achieving approximately 1.5x training efficiency and 5x data efficiency. In summary, our research makes three significant contributions to the field:

Table 1: Summary of recent innovations in file selection for LLM pre-training, categorized by their proxy model, proxy function to estimate sample importance, and whether requiring a target domain.

| Selection Method | Proxy Model ($M$) | Proxy Function ($F_M$) | Domain Independent |
|---|---|---|---|
| Heuristic Cls. | Trained binary classifier | Probability to target domain | ✗ |
| DSIR (NeurIPS'23) | Hashed n-gram extractor | Similarity to target distribution | ✗ |
| QuRating (ICML'24) | Trained judge model via GPT-3.5 | Judgement score of target ability | ✗ |
| INGENIOUS (Emnlp'23) | Pre-trained LLM with warming-up | Facility Location on similarity matrix | ✓ |
| D4 (NeurIPS'23) | Existing text feature extractor | Distance in feature space | ✓ |
| DiSF (Ours) | Existing text feature extractor | Decorrelation of feature dimensions | ✓ |

- We rethink recent file selection innovations in pre-training LLMs, and identify a diversity dilemma known as dimensional collapse in feature representation learning, improving performance in domain-specific tasks but causing severe degradation in overall performance (Section 2).
- To prevent collapse and enhance diversity, we propose DiSF, which selects the most decorrelated text files using a classical greedy algorithm, and analyze its approximation to the optimal solution under the formulation of $\gamma$-weakly submodular optimization problem (Section 3).
- We established a benchmark on TinyLlama architectures and SlimPajama text corpus with evaluation on nine tasks from Harness. Extensive experiments and ablations demonstrate the superior performance of our method, with improved training and sample efficiency (Section 4).

## 2 RETHINKING FILE SELECTION FOR LLM PRE-TRAINING UNDER BUDGET

In this section, we first introduce the definition of file selection objective, and then revisit recent file selection innovations for pre-training LLMs, with a particular focus on the diversity dilemma.

### 2.1 PROBLEM STATEMENT

Given training and selection budgets $\mathcal{T}$ and $\mathcal{S}$, our objective is to select the most valuable samples $\mathbb{V}$ from a text corpus $\mathbb{D} = \{D_1, ..., D_i, ..., D_N\}$ collected from various domains $D_i$ (e.g., Wikipedia or samples with high writing qualities) to optimize model weights $W$ on pre-training task $\mathcal{L}$, thereby maximizing general performance $\mathcal{A}$. While $\mathcal{A}$ is challenging to verify on a given LLM, it can be inferred through diverse abilities such as commonsense and problem-solving, with evaluation tools like Harness (Gao et al., 2024). Mathematically, $\mathbb{V}$ can be obtained through selection objective as

$$\arg\max_{U \subseteq \mathbb{D}} \ \mathcal{A}(\arg\min_{W} \mathcal{L}(W, \mathcal{T}, U)),$$
$$\text{s.t. } |U| \leq \mathcal{S}. \tag{1}$$

There are various options to define $\mathcal{T}$, $\mathcal{S}$, $\mathcal{A}$, and $\mathcal{L}$. In this work, we define the training budget $\mathcal{T}$ as the number of pre-trained tokens, the selection budget $\mathcal{S}$ as the number of files, the pre-training objective $\mathcal{L}$ as next-word prediction on SlimPajama (Computer, 2023; Touvron et al., 2023a), and the LLM's generic performance $\mathcal{A}$ as the evaluation across different tasks using the Harness framework.

### 2.2 RECENT SELECTION METHODS

Analyzing the objective defined in equation 1, the inner part minimizes the pre-training objective on LLM with a predefined training budget and the selected samples, while the upper level focuses on selecting the most valuable samples to maximize the LLM's generic performance within the selection budgets. Directly searching for valuable samples $\mathbb{V}$ in the full corpus $\mathbb{D}$ is extremely time-consuming and expensive. To reduce this cost, recent innovations in file selection for LLM pre-training mostly rely on using an existing or trained proxy model $M$ and designing a proxy function $F_M$ based on a target domain. Through linking text samples $x \in \mathbb{D}$ to the generic performance $\mathcal{A}$, they transform equation 1 into choosing the samples with the highest values of $F_M(x)$, as follows:

$$\mathbb{V} = \text{Top}_S F_M(x \in \mathbb{D}). \tag{2}$$

As summarized in Table 1, typical Heuristic classification (Brown, 2020b; Chowdhery et al., 2023b) trains a binary classifier to filter web data, selecting files with probabilities to a target format above a

noisy, such as BookCorpus and Wikipedia (Computer, 2023). Similarly, DSIR (Xie et al., 2023b) improves it and also treats these two domains as high-quality domains to select files for general purpose, with a hashed n-gram feature extractor to measure the similarity between the text features and the target distribution. QuRating (Wettig et al., 2024) queries GPT-3.5-turbo and trains a judge model to assess the text samples' quality of a target style, such as writing and mathematics. However, as shown in Figure 2, selection methods based on specific domains lead to a diversity dilemma, known as *dimensional collapse* in representation learning (Jing et al., 2022; Zbontar et al., 2021; Bardes et al., 2022; Chen et al., 2020), improving performance in the domain related task but causing severe degradation in overall performance across diverse tasks. The recent method D4 (Tirumala et al., 2023) and INGENIOUS (Renduchintala et al., 1991) reduce file redundancy by leveraging feature distances and selecting informative samples based on similarity kernels respectively, which improves diversity but can not effectively mitigate dimensional collapse as ours (see both Figure 1 and Figure 3).

## 2.3 DIVERSITY DILEMMA: DIMENSIONAL COLLAPSE

As shown in Figure 1, dimensional collapse occurs in the embedding space when samples are selected based on the target domains. Their embedding vectors extracted by the Contriever (Izacard et al., 2021) span a lower-dimensional subspace, indicating less diversity. To quantify this, we conduct an eigenvalue analysis on the covariance matrix of selected text features under different selection methods, and visualize the dominance score of the top$_k$ eigenvalues calculated as $\frac{\sum_{i=1}^{k} \lambda_i}{\sum_{j=1}^{d} \lambda_j}$, where $\lambda_i$ is the $i$-th large eigenvalue of the covariance matrix, and $d$ is the dimension of the feature space. Smaller dominance score suggests more uniform feature dimensions and richer information (Chen et al., 2020; Zbontar et al., 2021; Shi et al., 2022; Jing et al., 2022; Bardes et al., 2022). As demonstrated in Figure 3, Heuristic classification, DSIR, and QuRating with a focus on writing style (denoted as QuRating-W) exhibit significantly higher dominance scores compared to D4, INGENIOUS, and QuRating with all styles (denoted as QuRating-A), highlighting the reduced diversity caused by target domains selection criteria. To address this and enhance diversity, we propose a novel diversified file selection algorithm (DiSF) to select text samples, spanning more uniform feature dimensions in the embedding space. As red line shown in Figure 3, compared to D4

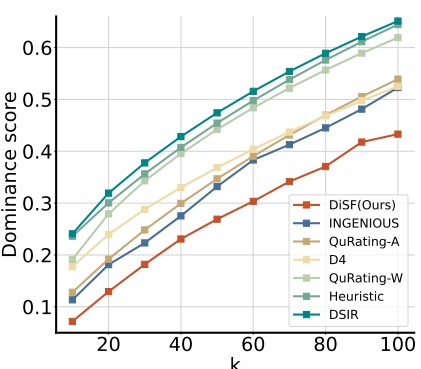

Figure 3: The dominance score for recent methods calculated as $\frac{\sum_{i=1}^{k} \lambda_i}{\sum_{j=1}^{d} \lambda_j}$, where $\lambda_i$ represents the $i$-th largest eigenvalue of the feature covariance matrix, and $d$ is the dimension of the feature space. We use Contriever model to extract features. We select the top 500 text samples based on their respective selection criteria. For D4, we select 500 random samples after reducing redundancy.

and QuRating-A, our DiSF further reduces the dominance score, demonstrating its effectiveness in uniforming dimensions and mitigating dimensional collapse.

## 3 DISF: DIVERSIFIED FILE SELECTION

In this section, we define the diversified selection criterion of DiSF, and the selection procedures with a classical greedy algorithm in batch level, and then analyze the function from view under $\gamma$-weakly submodular optimization theories, followed with time complexity analysis.

### 3.1 METHOD

**Selection criterion.** Given a set of $n$ text samples $U = \{x_1, ..., x_i, ...x_n\}$, their text features with a standard normalization $Z = \{z_1, ..., z_i, ...z_n\}$ are obtained by a text feature extractor $M$ as

$$z_i = \frac{f(x_i, M) - \mu}{\sigma},$$
(3)

where $f$ calculates the feature representations of text samples, and $\mu$ and $\sigma$ are the mean and variance of $\{f(x_1, M), ..., f(x_i, M), ...f(x_n, M)\}$, respectively. Then, the covariance matrix C is defined as

$$C(U, M) = \frac{1}{n-1} \sum_{i=1}^{n} z_i^T z_i. \tag{4}$$

As discussed in Section 2, our goal is to prevent dimensional collapse by ensuring more uniform eigenvalues in the covariance matrix of the selected samples. Directly calculating and selecting based on eigenvalues are costly, but it is feasible to optimize the Frobenius norm of the covariance matrix $\|C\|_F$ (Zbontar et al., 2021; Bardes et al., 2022; Shi et al., 2022), as described in Lemma 1:

**Lemma 1.** *Assuming a covariance matrix $C \in \mathbf{R}^{d \times d}$ computed from the features with the standard normalization, and its eigenvalues $\{\lambda_1, \lambda_2, ..., \lambda_d\}$, we will have the following equality that satisfied*

$$\sum_{i=1}^{d} (\lambda_i - \frac{1}{d} \sum_{j=1}^{d} \lambda_j)^2 = \|C\|_F^2 - d.$$

We provide a detailed proof in Appendix A.6 for clarity. From Lemma 1, it is evident that ensuring the uniformity of the eigenvalues of the covariance matrix can be translated into minimizing the Frobenius norm of the covariance matrix. Thus, given any text set $U \subseteq \mathbb{D}$, we evaluate its importance as $-\|C(U, M)\|_F$, and define the selection objective as

$$\arg\max_{U} -\|C(U, M)\|_F, \quad s.t. \ |U| \leq \mathcal{S}, \tag{5}$$

where $\mathcal{S}$ is the predefined selection budget. Differing from typical selection algorithms that directly define the importance of individual text samples through proxy function shown in equation 1, our selection objective defined in equation 5 is calculated on a subset of samples.

**Selection Procedure.** To satisfy the non-negative requirement in the later analysis, we first reformulate our proxy function into a non-negative form as

$$F_M^{DiSF}(U) = e^{-\|C(U,M)\|_F}, \tag{6}$$

where $U \subseteq \mathbb{D}$. We apply the classical greedy algorithm to select the most valuable samples based on our proxy function. This allows us to iteratively choose the most valuable samples as follows:

$$U \leftarrow U \cup \{\arg\max_{x \in \mathbb{D} \setminus U} F_M^{DiSF}(U \cup \{x\})\}. \tag{7}$$

However, directly applying the greedy algorithm as shown in equation 7 to the entire text corpus is computationally expensive, we perform the selection at the batch scale. In Section 4.4, we provide an abla-

---

**Algorithm 1** Selection procedure of DiSF

**Input:** $(\mathbb{D}, b, \mathcal{S}, M)$

$\quad \mathbb{V} \leftarrow \emptyset$.
$\quad$ Divide $\mathbb{D}$ into batches of $b_i$ with scale $b$.
$\quad$ **for** $i = 1, \ldots, \lfloor \frac{|\mathbb{D}|}{b} \rfloor$ **do**
$\quad\quad$ randomly select $x^* \in b_i$ and $U_i \leftarrow \{x^*\}$.
$\quad\quad$ **while** $|U_i| \leq \frac{b|\mathcal{S}|}{|\mathcal{D}|}$ **do**
$\quad\quad\quad$ $b_i \leftarrow b_i \setminus \{x^*\}$.
$\quad\quad\quad$ $x^* = \arg\max_{x \in b_i} F_M^{DiSF}(U_i \cup \{x\})$.
$\quad\quad\quad$ $U_i \leftarrow U_i \cup \{x^*\}$.
$\quad\quad$ **end while**
$\quad\quad$ $\mathbb{V} = \mathbb{V} \cup U_i$.
$\quad$ **end for**

**Output:** $\mathbb{V}$

---

tion study of selection scale. Given a text corpus $\mathbb{D}$, selection scale $b$, selection budget $S$, and proxy model $M$, the selection process is outlined in Algorithm 1. We first divide the text corpus into $\lfloor \frac{|D|}{b} \rfloor$ batches, where $\lfloor \cdot \rfloor$ is the round down command. In each batch $b_i$, we initialize $U_i$ with a randomly selected sample and remove it from $b_i$. Then, we iteratively apply $(\lfloor \frac{b|\mathcal{S}|}{|\mathbb{D}|} \rfloor - 1)$ times the greedy algorithm, removing the most valuable sample from $b_i$ and adding it to $U_i$ based on equation 7. Finally, the selected samples are the combination of all $U_i$.

### 3.2 ANALYSIS

**Selection Analysis.** Diversity is often effectively modeled by submodular functions (Nemhauser et al., 1978; Fujishige, 2005; Balakrishnan et al., 2022; Hong et al., 2024; Zhang et al., 2025; 2024b; 2023), which exhibit a property of diminishing returns, i.e., the marginal gain an element brings to a subset decreases as more elements are added to that subset. Mathematically, given a set $\Omega$, a set function $f : 2^N \rightarrow \mathbb{R}$ is $\gamma$-weakly submodular if and only if, for any subsets $A \subseteq B \subseteq \Omega$, and a element $x \in \Omega \setminus B$, the following inequality holds:

$$f(A \cup \{x\}) - f(A) \geq \gamma(f(B \cup \{x\}) - f(B)), \tag{8}$$

where $\gamma \in (0,1]$.

A non-negative monotone $\gamma$-weakly submodular maximization problem can be solved using the classical greedy algorithm, which guarantees a $(1-e^{-\gamma})$-approximation to the optimal solution (function value on selected samples compared to optimal value) (DAS, 2011). As demonstrated in Figure 4, we empirically verify the monotonicity of our proxy function by increasing the number of samples randomly chosen from the selected text files of different methods. Additionally, the results in Figure 4 indicate a significantly higher proxy value for our selection method compared to others, demonstrating the superior ability to mitigate dimensional collapse. Since our proxy function also aims to capture the diversity within a given set, we try to provide some insights into how our algorithm approaches the optimal solution under this formulation. To provide a theoretical guarantee of the selection, we aim to establish a lower bound for the submodular ratio under any given set where the gain is positive, as most empirical verifications in Figure 4 suggest. Since the original formulation is challenging to analyze directly, we introduce two bounds on the function value. One is the upper bound $\epsilon$ on gain difference, as demonstrated in Assumption 1, and the other one is the upper bound $\mu$ on the average utility of function value as shown in Assumption 2. Therefore,

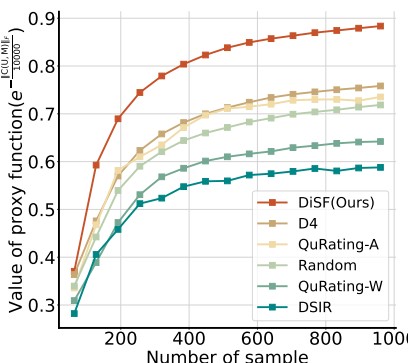

Figure 4: Proxy value, as defined in equation 6, calculated on different methods. For each method, we randomly choose samples from their selected text files with 1.5% selection budget. All cases generally demonstrate the property of monotonicity. Moreover, our selection method achieves significantly larger proxy values, indicating much better uniformity of feature dimensions.

we prove a lower bound of the submodular ratio of $e^{-2\mu}\frac{e^{2\mu-\epsilon}-1}{e^{2\mu}-1}$, which means given assumptions 1, 2, and positive gain, our function is at least $e^{-2\mu}\frac{e^{2\mu-\epsilon}-1}{e^{2\mu}-1}$-weakly submodular function. The detailed proof is provided in the Appendix A.5.3

**Time Complexity Analysis.** Given the feature dimension $d$, the time complexity of our DiSF is less than $O(|\mathcal{S}|^2bd^2)$, and we provide a detailed calculation of the results. All terms in the time complexity are at most quadratic and independent of the overall dataset size, which we consider acceptable for applying to file selection for LLM pre-training. Additionally, in Figure 7 of Section 4.4, we report the exact time required to select data on the benchmark dataset. In Section 4.3 and Appendix A.5.2, we compare the sampling and training efficiency of our method against the baselines.

## 4 EXPERIMENT

This part introduces the experimental setup, including dataset, evaluations, model architecture, baselines, and training details in Section 4.1, the main results and efficiency analysis in Section 4.2 and Section 4.3, and extensive ablation studies for better understanding of our DiSF in Section 4.4.

### 4.1 SETUP

**Dataset and evaluation.** Following many prior works (Touvron et al., 2023a; Zhang et al., 2024a; Wettig et al., 2024; Xie et al., 2023a), we employ **SlimPajama** (Touvron et al., 2023a; Computer, 2023) as the text corpus, which is specifically curated for pre-training LLMs. All selections are performed on about 590M training files of SlimPajama, processed with Llama tokenizer (Touvron et al., 2023a). Notably, QuRating (Wettig et al., 2024) provides judgments on various properties of text samples in SlimPajama using a judge model trained based on GPT-3.5-turbo, which can be directly utilized in our implement. To capture the diversity dilemma and evaluate generic performance of pre-trained LLMs, we use seven commonsense reasoning tasks from the popular framework **Harness** (Gao et al., 2024), including four reading comprehension tasks (**ARC-e**, **ARC-c** (Clark et al., 2018), **OBQA** (Mihaylov et al., 2018), and **BoolQ** (Clark et al., 2019)), and three physical world knowledge tasks (**PIQA** (Bisk et al., 2020), **HellaSwag** (Zellers et al., 2019), and **WinoGrande** (Sakaguchi et al., 2021)). Besides, we also employ another two tasks **MMLU** (Hendrycks et al., 2021), and **BBH** (Suzgun et al., 2022) from Harness to evaluate the problem-solving capabilities for further clarify of our effectiveness. See Section A.1 for detailed introduction of the dataset and tasks.

Table 2: Performance on seven tasks and their average with the Harness framework. In the upper part, we pre-train TinyLlama with 120M, 560M, and 1.1B parameters from scratch, with training budget of 10B tokens and selection budget of 1.5% of total training files. In the lower part, the training budget is increased to 50B. Results impacted by the diversity dilemma, best individual task results, and the best average performance are respectively highlighted in bold blue, black, and red.

| Model Size | Method | Pre-training TinyLlama from scratch with 10B training budget on 1.5% selected files | | | | | | | |
| --- | --- | --- | --- | --- | --- | --- | --- | --- | --- |
| | #Metric | ARC-e | ARC-c | OBQA | BoolQ | WinoGrande | HellaSwag | PIQA | Avg. |
| 120M | Random | 37.1 | 18.2 | 12.8 | 61.1 | 49.9 | 27.0 | 58.7 | 37.8 |
| | DSIR | 36.6 | 18.2 | 13.8 | **57.6** | 51.7 | 25.1 | **54.8** | 36.9 |
| | D4 | 38.0 | 18.1 | 13.6 | 60.1 | 51.8 | **27.8** | 57.8 | 38.2 |
| | QuRating-W | 37.7 | **18.6** | **15.0** | 60.9 | 51.3 | **24.3** | **55.6** | 37.7 |
| | QuRating-A | 39.1 | 18.3 | 14.2 | 60.7 | 50.8 | 27.4 | 58.4 | 38.4 |
| | Doremi | 36.8 | 18.2 | 13.8 | 60.9 | **52.7** | 27.3 | 58.8 | 38.5 |
| | INGENIOUS | 39.5 | 18.6 | 13.4 | 60.2 | 50.4 | 27.4 | 58.7 | 38.3 |
| | DiSF (Ours) | **39.9** | 17.8 | 13.8 | **61.7** | 51.9 | 27.6 | **59.5** | **38.9** |
| 560M | Random | 43.2 | 19.9 | 15.2 | 59.3 | 52.9 | 30.4 | 62.0 | 40.4 |
| | DSIR | 41.8 | 19.3 | 16.8 | 60.5 | 50.3 | **26.0** | **56.0** | 38.7 |
| | D4 | 46.5 | 19.5 | 16.0 | 60.5 | **53.2** | 29.5 | 61.4 | 40.9 |
| | QuRating-W | 42.0 | **21.3** | 17.4 | 59.5 | **49.8** | 28.4 | **58.2** | 39.5 |
| | QuRating-A | 46.7 | 19.1 | 17.2 | **61.3** | 51.1 | 28.9 | 63.4 | 41.1 |
| | INGENIOUS | 45.8 | 20.6 | 16.0 | 58.3 | 51.4 | 30.8 | 62.5 | 40.8 |
| | Doremi | 42.7 | 19.3 | 15.8 | 60.9 | 50.4 | 30.0 | 63.7 | 40.4 |
| | DiSF (Ours) | **47.5** | 21.2 | 16.2 | 58.9 | 51.0 | **31.1** | **64.2** | **41.4** |
| 1.1B | Random | 44.8 | 19.0 | 16.4 | 59.9 | 51.3 | 30.8 | 64.1 | 40.9 |
| | DSIR | 45.7 | 20.3 | 18.6 | 59.8 | 50.4 | **27.6** | **58.3** | 40.1 |
| | D4 | 46.2 | 19.3 | 18.8 | 60.2 | 51.3 | 30.9 | 65.4 | 41.7 |
| | QuRating-W | 44.4 | **21.4** | 17.0 | 59.6 | 51.4 | 31.0 | **60.1** | 40.7 |
| | QuRating-A | 47.4 | 20.9 | **19.8** | 59.1 | 50.2 | 30.1 | 63.3 | 41.6 |
| | INGENIOUS | 45.3 | 19.6 | 19.8 | 60.0 | 51.2 | 31.2 | 64.9 | 41.7 |
| | Doremi | 44.7 | 19.7 | 17.6 | **61.0** | 51.2 | 31.1 | 64.6 | 41.4 |
| | DiSF (Ours) | **47.7** | 19.5 | 18.2 | 59.7 | **52.2** | **32.3** | **65.6** | **42.2** |
| Model Size | Method | Pre-training TinyLlama from scratch with 50B training budget on 1.5% selected files | | | | | | | |
| | #Metric | ARC-e | ARC-c | OBQA | BoolQ | WinoGrande | HellaSwag | PIQA | Avg. |
| 120M | Random | 41.5 | 17.6 | 16.6 | 56.3 | 52.2 | 28.4 | 59.5 | 38.9 |
| | DSIR | 44.2 | 18.4 | 17.2 | **53.9** | **49.6** | **25.2** | **56.3** | 37.8 |
| | D4 | 42.1 | 19.2 | 17.0 | 58.3 | 52.6 | 28.1 | 60.9 | 39.7 |
| | QuRating-W | 41.6 | 18.9 | 16.2 | 59.5 | 51.6 | 27.6 | **56.8** | 38.9 |
| | QuRating-A | **46.8** | **19.7** | 15.8 | 60.5 | 51.1 | 27.9 | 58.4 | 40.0 |
| | INGENIOUS | 42.2 | 19.1 | 16.3 | 58.7 | 51.2 | 28.5 | 61.0 | 39.6 |
| | Doremi | 40.7 | 18.9 | 16.2 | 60.2 | 52.7 | 28.1 | **61.4** | 39.7 |
| | DiSF (Ours) | 44.3 | 18.3 | **17.8** | **61.1** | **53.3** | 28.7 | 61.0 | **40.6** |
| 560M | Random | 46.0 | 20.9 | 16.8 | 59.0 | 52.5 | 31.6 | 64.7 | 41.6 |
| | DSIR | 45.9 | 22.2 | 18.5 | 58.1 | 50.7 | **27.8** | **59.3** | 40.4 |
| | D4 | 47.2 | 21.7 | 18.2 | 58.7 | 52.2 | 32.4 | 65.3 | 42.2 |
| | QuRating-W | 46.9 | **23.4** | **19.5** | **56.5** | 52.2 | **28.5** | **61.3** | 41.2 |
| | QuRating-A | **48.3** | 19.2 | 18.2 | 58.9 | 52.1 | **34.1** | **67.4** | 42.6 |
| | INGENIOUS | 46.8 | 21.6 | 17.0 | 59.3 | 52.2 | 32.8 | 66.5 | 42.3 |
| | Doremi | 47.2 | 22.4 | 18.6 | 59.0 | 52.6 | 33.1 | 66.2 | 42.7 |
| | DiSF (Ours) | 47.3 | 22.0 | 18.8 | **60.8** | 52.8 | 33.5 | **67.4** | **43.2** |
| 1.1B | Random | 51.4 | 20.9 | 18.2 | 56.2 | 51.3 | 34.7 | 67.3 | 42.9 |
| | DSIR | 50.2 | 20.6 | 20.0 | 54.6 | 52.4 | **30.4** | **64.2** | 41.8 |
| | D4 | 53.6 | 22.1 | 20.9 | 57.3 | 52.9 | 35.2 | 67.7 | 44.2 |
| | QuRating-W | **53.9** | 23.1 | 20.8 | 55.0 | 52.7 | 35.0 | **63.3** | 43.4 |
| | QuRating-A | 53.6 | **23.7** | 21.2 | 60.1 | 51.6 | 36.5 | 67.0 | 44.8 |
| | INGENIOUS | 51.7 | 21.6 | 22.2 | 56.9 | 51.7 | 36.8 | 67.9 | 44.1 |
| | Doremi | 51.2 | 21.9 | **21.8** | 58.1 | **54.3** | 36.6 | 67.8 | 44.5 |
| | DiSF (Ours) | 51.8 | 22.7 | 20.0 | **62.0** | 53.5 | **37.3** | **69.0** | **45.2** |

**Model architecture.** As for the model, we adopt **Tinyllama** architecture (Zhang et al., 2024a) with 120M, 560M, and 1.1B parameters. Thanks to FlashAttention (Dao et al., 2022) and Lit-GPT (LightningAI, 2023), all experiments can be conducted on NVIDIA GeForce RTX 4090 GPUs with 24GB memory, which is feasible for general academic research. All experiments and selection are implemented by PyTorch (Paszke et al., 2019) on platforms with 8 GPUs and 64 CPUs. Except for Tinyllama, we also adopt OPT (Zhang et al., 2022) and Pythia (Biderman et al., 2023) to verify the scalability of our method on model architecture as ablation study shown in Section 4.4. For feature extraction, we utilize the Contriever (Izacard et al., 2021) with approximately 110M parameters to calculate feature representations of the text samples as defined in equation 3. We also experiment

with other pre-trained models as feature extractors, including the text encoder of CLIP (Radford et al., 2021) and GPT-2 (Radford et al., 2019), as discussed in the ablation study of Section 4.4. See Appendix A.4 for detailed introduction of model structures, as well as their training times.

**Baselines.** We compare our DiSF with **Random** selection and existing file selection methods for LLM pre-training, including **DSIR** (Xie et al., 2023b), QuRating (Wettig et al., 2024), **INGE-NIOUS** (Renduchintala et al., 1991), and **D4** (Tirumala et al., 2023). Since DSIR improves on Heuristic classification (Brown, 2020b; Chowdhery et al., 2023b), we present only DSIR results based on Wikipedia and Books. For QuRating, based on the top judgment values, we select text samples for writing style (denoted as **QuRating-W**) and uniformly select across all styles (denoted as **QuRating-A**). For INGENIOUS, we utilize Contriever to extract features, which is more efficient than warmed-up model, as analyzed in Appendix A.3.4. Additionally, for a comprehensive comparison, we include **Doremi** (Xie et al., 2023a), a recently proposed method that produces weights for pre-training on multiple text domains. We use the weights as the selection ratio of text samples in different domains in our experiment. Notably, in our ablation studies, we also compare our method with **Full Data** pre-training, which refers to pre-training the LLMs on all training files in SlimPajama until the specified training budget is reached. See Appendix A.3 for detailed implementation of the baselines.

**Pre-training details.** We follow all settings in TinyLlama (Zhang et al., 2024a). The optimizer is AdamW (Loshchilov & Hutter, 2019), setting parameters $\beta_1$ at 0.9 and $\beta_2$ at 0.95. We adopt the cosine learning rate schedule with a maximum learning rate of 4e-4 and the minimum of 4e-5, the batch size of 2M tokens, the weight decay of 0.1, and the gradient clipping threshold of 1. The training budgets are 10B and 50B tokens, with 1.5% selection budget of SlimPajama's training files. Note that, we choose to report performance with a 1.5% selection budget, since it achieves comparable performance compared to Full Data pre-training under 50B pre-training budget on TinyLlama 1.1B. Unless otherwise specified, the selection scale $b$ is set to 1024. See Appendix A.4 for more details.

## 4.2 Main Results

**Performance on commonsense reasoning tasks.** As shown in Table 2, selection methods based on a target domain such as DSIR, D4, and QuRating-W improve performance on reading comprehension tasks like ARC-c and OBQA, but suffer significant declines on physical world knowledge tasks especially HellaSwag and PIQA (highlighted in blue), revealing the diversity dilemma. In contrast, Doremi, D4, QuRating-A, and our DiSF, which select samples from multiple text domains, achieve competitive results on OBQA and ARC-c while significantly improving overall performance across the remaining tasks. This highlights the critical importance of diversity in pre-training LLMs. Notably, our DiSF outperforms all baselines in terms of average performance across the seven tasks, with an average improvement of 2.5% compared to DSIR. Furthermore, with increasing training budget, the improvement on DSIR becomes more pronounced, rising from 2.1% to 3.4% on TinyLlama 1.1B.

**Scalability on model size.** To verify the scalability of selection methods across different model sizes, we use the TinyLlama architecture with models of 120M, 560M, and 1.1B parameters. As shown in Table 2, our DiSF consistently outperforms all baselines, demonstrating strong scalability with increasing model size. Notably, as the model scales from 120M to 1.1B, DiSF's improvement over DSIR grows from 2.8% to 3.4%, suggesting even greater effectiveness for larger LLMs.

**Performance on problem-solving tasks.** As shown in Table 3, we further verify the performance of pre-trained TinyLlama 1.1B on two additional problem-solving tasks,

Table 3: Problem-solving performance on MMLU (5 shot) and BBH (3 shot) of TinyLlama 1.1B with 1.5% selection budget and 50B pre-training budget.

| Method | MMLU(5 shot) | BBH(3 shot) |
|---|---|---|
| DSIR | $22.9_{\pm 0.4}$ | $18.5_{\pm 0.4}$ |
| QuRating-W | $23.1_{\pm 0.2}$ | $18.7_{\pm 0.4}$ |
| QuRating-A | $24.5_{\pm 0.4}$ | $20.2_{\pm 0.4}$ |
| D4 | $24.3_{\pm 0.5}$ | $20.8_{\pm 0.4}$ |
| Doremi | $23.1_{\pm 0.5}$ | $20.3_{\pm 0.4}$ |
| DiSF (Ours) | $\mathbf{25.4}_{\pm 0.4}$ | $\mathbf{21.2}_{\pm 0.4}$ |

MMLU and BBH, using various selection methods. Except for better commonsense abilities, results of our DiSF shown in Table 3, demonstrate a better problem-solving capability, outperforming all other baselines. Specifically, on MMLU and BBH, our DiSF respectively achieves 2.5% and 2.7% improvement compared to DSIR.

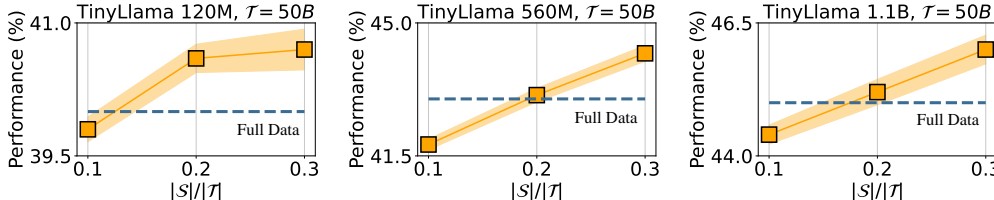

Figure 5: Average performance of TinyLlama pre-trained after 50B tokens on files selected by our DiSF with varying selection budgets compared to Full Data pre-training on SlimPajama.

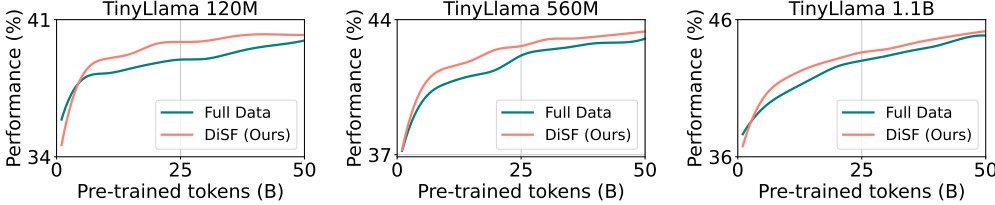

Figure 6: Performance of TinyLlama pre-trained under our DiSF with 1.5% selection budget compared to Full Data pre-training on SlimPajama, as the pre-trained tokens (training budget) increases.

## 4.3 EFFICIENCY ANALYSIS

In this section, we try to analyze how many samples and computations we can save within the training budget, compared to Full Data pre-training, that is *data efficiency* and *training efficiency*. Please note that, Full Data pre-training refers to pre-training the LLMs on all SlimPajama's training files until the specified training budget is reached. In Figure 5, we demonstrate the average commonsense performance of TinyLlama with 120M, 560M, and 1.1B parameters, pre-trained after 50B tokens on files selected by DiSF with varying selection budgets compared to Full Data. It is evident that to achieve the equivalent or superior performance compared to Full Data with 50B training budget, we only need to select about 20% of the pre-trained tokens (1.5% of SlimPajama's training files), achieving at least *5x data efficiency*. In Figure 6, we show the curves of averaged common-

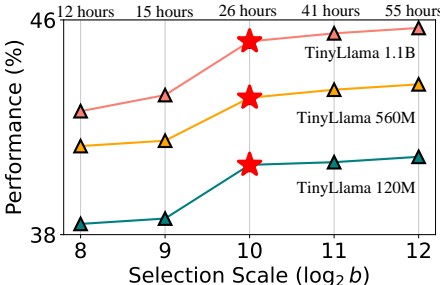

Figure 7: Performance of pre-trained TinyL-lama under different selection scales ($\log_2 b$). With 1.5% selection budget and 50B training budget, we identify an ideal point with acceptable computational cost and near-optimal performance, marked by a red star.

sense reasoning performance during the pre-training under our DiSF with 1.5% selection budget compared to Full Data. It can be seen that, when achieving the performance of Full Data pre-trained with 50B tokens, our methods only need 27B, 36B, and 40B tokens, respectively on TinyLlama with 120M, 560M, and 1.1B parameters, achieving an average of *1.5x training efficiency*.

## 4.4 ABLATION STUDY

**Selection budget.** This part analyzes the performance of pre-trained LLM with different selection budgets when using our DiSF. We conduct the ablation on TinyLlama 120M with 50B training budget and present its average commonsense reasoning performance. As shown in Figure 8, the performance initially rises rapidly with increasing selection ratios, reaching a peak of 41.2 with 3% of total training files (about 20B tokens). Once the selected samples exceed 3%, the performance begins to decline and converge to the performance of Full Data pre-training. This insight may help choose selection ratios and enhance understanding of our selection algorithm.

Table 4: Ablation study of our DiSF with different model architectures compared to other selection methods, respectively using TinyLlama with 1.1B parameters, Pythia with 1B parameters and OPT with 1.3B parameters.

| Method | TinyLlama | Pythia | OPT |
|---|---|---|---|
| DSIR | 41.8 | 41.4 | 43.1 |
| D4 | 44.2 | 43.9 | 44.7 |
| QuRating-A | 44.8 | 43.6 | 45.1 |
| Doremi | 44.5 | 43.7 | 44.9 |
| Ours | **45.2** | **44.2** | **45.5** |

**Model architecture.** To further verify performance across different architectures, we additionally adopt two large language models: Pythia (Biderman et al., 2023) with 1B parameters and OPT (Zhang et al., 2022) with 1.3B parameters. As shown in Table 4, we compare the average commonsense reasoning performance of our DiSF with baselines, including DSIR, D4, QuRating-A, and Doremi, under the same training budget of 50B pre-trained tokens. The results demonstrate that our DiSF consistently outperforms the baselines, highlighting its effectiveness and scalability across different model architectures.

**Feature extractor.** In our experiments, we utilize a proxy model named Contriever (Izacard et al., 2021), with about 110M parameters, as feature extractor to provide the embedding space for selected text samples. We also try two other pre-trained models: the text encoder of CLIP (Radford et al., 2021) with about 70M parameters and GPT-2 (Radford et al., 2019) with about 117M parameters. As shown in Table 5, CLIP fails to deliver satisfactory performance, because it struggles with longer text sequences. GPT-2 does not outperform the Contriever model because

Table 5: Ablation study of our DiSF with different feature extractor. We show results, respectively using Contriever with 110M parameters, text encoder of CLIP with 70M parameters and GPT-2 with 117M parameters.

| TinyLlama | CLIP | Contriever | GPT-2 |
|---|---|---|---|
| 120 M | 39.5 | **40.6** | 40.0 |
| 560 M | 42.1 | **43.1** | 42.3 |
| 1.1 B | 43.9 | **45.2** | 44.4 |

it is optimized for autoregressive text generation rather than producing meaningful sentence representations, whereas Contriever, specifically designed for measuring text similarity, excels at capturing text diversity.

**Selection scale.** To manage the computational cost of file selection, we apply submodular optimization with classical greedy algorithm at the batch scale, as shown in Algorithm 1, introducing the hyper-parameter, selection scale $b$. The computational complexity of the selection process, defined in equation 7, is $\mathcal{O}(\frac{b}{|\mathcal{S}|^2})$, where the larger $b$ increases the computational burden. As ablation on $b$ shown in Figure 7, a larger batch size significantly raises computational costs, while a smaller batch size fails to select sufficiently diversified files. With a fixed selection budget of 1.5%, training budget of 50B tokens, and Contriever as the feature extractor, we identified an ideal point of 1024, marked by a red star, which balances computational time and near-optimal performance. This insight may provide valuable guidance for choosing an appropriate selection scale in other settings.

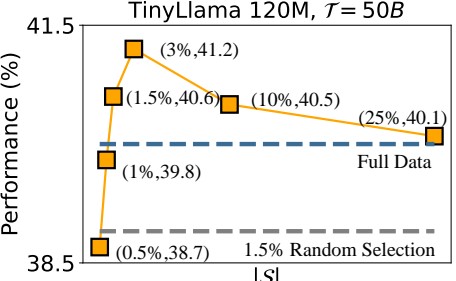

Figure 8: Average commonsense reasoning performance of TinyLlama 120M pre-trained under DiSF with different selection budget and 50B training budget. Blue and grey lines respectively denote Full Data pre-training and random selection with 1.5% selection budget.

## 5 CONCLUSION

In this work, we revisit recent innovations in file selection for pre-training large language models (LLMs) and identify a diversity dilemma: dimensional collapse, where performance improves on specific tasks but degrades overall across diverse tasks. To address this, we propose a novel Diversified File selection method (DiSF) which selects decorrelated text files in the embedding space to enhance diversity. DiSF achieves more uniform eigenvalues of the feature covariance matrix by minimizing its Frobenius norm and solve it with a greedy algorithm. We analyze its time complexity and approximation to optimal solution under $\gamma$-weakly submodular optimization, and establish a benchmark with TinyLlama architecture, evaluating performance across nine tasks from the Harness framework. Extensive experiments and ablation studies demonstrate the critical role of diversity in file selection for LLM pre-training and showcase DiSF's superior effectiveness and efficiency.

## 6 ACKNOWLEDGEMENT

The work is supported by the National Key R&D Program of China (No. 2022ZD0160702), STCSM (No. 22511105700, No. 21DZ1100100), 111 plan (No. BP0719010), National Natural Science

Foundation of China (No. 62306178), Shenzhen Basic Research Project (Natural Science Foundation) Basic Research Key Project (NO. JCYJ20241202124430041), CCF-Baidu Open fund (NO. CCF-Baidu202413), and the National Research Foundation, Singapore, under its NRF Professorship Award No. NRF-P2024-001. Ziqing Fan is partially supported by Wu Wen Jun Honorary Doctoral Scholarship, AI Institute, Shanghai Jiao Tong University.

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

# A    APPENDIX

In the Appendix, we provide additional information including dataset, baselines, model and training details, and the proof of Lemma 1. As shown in the following, to help readers read, we provide a table of contents of the full paper.

## CONTENTS

## A.1   RELATED WORK

**Efficient File Selection**   Pre-trained Large Language Models (LLMs) have exhibited exceptional performance across a wide range of downstream tasks (Jiang et al., 2025; Ye et al., 2024; Pang et al., 2024; Yi et al., 2024; Wang et al., 2024; Lee et al., 2021; Tang et al., 2024). However, their training process is computationally intensive, with costs escalating significantly as both model size and training data volume increase (Rae et al., 2021; Patterson et al., 2021; Thoppilan et al., 2022). To maximize the performance of LLMs within constrained budgets, it is crucial to curate high-quality pre-training data from extensive text corpora, as this enhances both training efficiency and sample effectiveness. Recent advancements in pre-training file selection for LLMs primarily rely on leveraging existing or trained proxy models, coupled with proxy functions designed to assess the similarity of data to a target domain, which is typically considered high-quality. For instance, heuristic classification methods (Brown, 2020b; Chowdhery et al., 2023b) employ binary classifiers to filter content resembling text from domains such as Books and Wikipedia (Computer, 2023). Similarly, DSIR (Xie et al., 2023b) focuses on these domains by utilizing a hashed n-gram extractor to quantify similarity. Another approach, QuRating (Wettig et al., 2024), utilizes GPT-3.5-turbo to train a judge model that evaluates the quality of domains like writing and education. Despite these innovations, methods anchored to specific domains face a diversity dilemma, akin to the phenomenon of *dimensional collapse* observed in representation learning (Jing et al., 2022; Zbontar et al., 2021; Bardes et al., 2022; Shi et al., 2022; Fan et al., 2024). These approaches often result in feature vectors that span a lower-dimensional subspace, reflecting reduced diversity. While this may enhance performance in domain-specific tasks such as reading comprehension, it leads to significant performance degradation across broader, more diverse domains, particularly in tasks requiring physical world knowledge, such as PIQA (Bisk et al., 2020) and HellaSwag (Zellers et al., 2019). To tackle this issue and promote greater diversity, we introduce a novel diversified file selection algorithm, termed DiSF, designed to curate text samples that span a more uniform distribution of feature dimensions within the embedding space.

**Submodular optimization.**   Submodularity is a property of functions defined on a set $\Omega$ that exhibit diminishing returns. Submodular functions like facility location, log determinant, and graph cut (Salhi, 1991b; Fujishige, 2005; Krause & Golovin, 2014; Kaushal et al., 2019; Karanam et al., 2022) are widely recognized for effectively modeling diversity (Nemhauser et al., 1978; Wei et al., 2015; Balakrishnan et al., 2022; Hong et al., 2024). Moreover, submodular optimization has achieved significant success in various fields, such as summarization (Kothawade et al., 2022; Kumari et al., 2024), curriculum learning (Balakrishnan et al., 2022; Zhou & Bilmes, 2018), active learning (Wei et al., 2015; Guillory & Bilmes, 2011), and subset selection (Jain et al., 2024; Lin & Bilmes, 2011), where selecting diverse subsets is crucial. Given a set $\Omega$, a set function $f : 2^N \to \mathbb{R}$ is $\gamma$-weakly submodular if and only if, for any subsets $A \subseteq B \subseteq \Omega$, and a element $x \in \Omega \setminus B$, the following inequality holds:

$$f(A \cup \{x\}) - f(A) \geq \gamma(f(B \cup \{x\}) - f(B)), \tag{9}$$

where $\gamma \in (0,1]$. When $\gamma = 1$ (Santiago & Yoshida, 2020; DAS, 2011), the function is submodular function. A non-negative monotone $\gamma$-weakly submodular maximization problem can be solved using the classical greedy algorithm, which guarantees a $(1 - e^{-\gamma})$-approximation. In this work, our function aims to evaluate diversity, making it well suited to be verified under this formulation.

## A.2   DATASET: SLIMPAJAMA

SlimPajama is a high-quality text corpus specifically created for pre-training large language models. This corpus, derived from RedPajama (Computer, 2023), underwent additional cleaning and deduplication processes. The original RedPajama corpus, an open-source research project, was designed to replicate the pretraining data of Llama (Touvron et al., 2023a) and contains over 1.2

trillion tokens. After extensive filtering to remove low-quality and duplicate content, SlimPajama retains only 50% of the original tokens from RedPajama. Notably, compared to domains like ArXiv, GitHub, and StackExchange, Heuristic classification and DSIR tend to select a higher proportion of files from Books and Wikipedia, and Qurating-W also favors the Books domain. This bias toward specific domains may explain why these methods encounter dimensional collapse and performance degradation in overall task performance.

### A.2.1 EVALUATION

**Harness evaluation framework.** To demonstrate the diversity dilemma and evaluate the general performance of the pre-trained LLMs, we utilize seven commonsense reasoning tasks from the widely recognized evaluation framework, Harness (Gao et al., 2024), including four reading comprehension tasks: **OBQA** (Mihaylov et al., 2018): Inspired by open book exams, this task tests the ability to comprehend

Table 6: Commonsense performance of IN-GENIOUS when using Warmed-up model (INGENIOUS-W) and Contriever (INGENIOUS-C) with 50B training budget.

| Method | TinyLlama-120M | TinyLlama-1.1B |
|---|---|---|
| INGENIOUS-W | 39.0 | 39.6 |
| INGENIOUS-C | 43.2 | 44.1 |

and apply knowledge similarly to human understanding; **ARC-e and ARC-c** (Clark et al., 2018): 7,787 multiple-choice science questions at a grade-school level, divided into an easy set (ARC-e) and a challenge set (ARC-c); **BoolQ** (Clark et al., 2019): A reading comprehension task that focuses on naturally occurring yes/no questions, and three physical world knowledge tasks: **HellaSwag** (Zellers et al., 2019): A collection to assess physically situated commonsense reasoning capabilities; **WinoGrande** (Sakaguchi et al., 2021): An expansion of the Winograd Schema Challenge (WSC) with increased scale and complexity; **PIQA** (Bisk et al., 2020): A task to measure the understanding and reasoning about physical interactions in the real world. To further verify our effectiveness, we also employ two problem-solving tasks from Harness: **MMLU** (Hendrycks et al., 2021): A task to measure the world knowledge and problem-solving capabilities across various subjects; **BBH** (Suzgun et al., 2022): A subset of 23 challenging tasks from the BIG-Bench benchmark (Srivastava et al., 2022) to measure the ability of complex instruction following.

Table 7: Commonsense performance when pre-trained on SlimPajama with Open-Llama 3B, 1.5% selection ratio and 10B training budget.

| Method | Random | DSIR | QuRating-W | QuRating-A | DiSF (Ours) |
|---|---|---|---|---|---|
| Commonsense ability | 43.1 | 41.9 | 42.6 | 43.6 | 44.4 |

Table 8: Commonsense performance when pre-trained on both StarcoderData, and SlimPajama with TinyLlama 1.1B, 1.5% selection ratio and 10B training budget.

| Method | Random | DSIR | DiSF (Ours) |
|---|---|---|---|
| Commonsense ability | 40.3 | 40.0 | **41.4** |
| bbh | 12.6 | 12.5 | **13.3** |
| mmlu | 23.0 | 22.1 | **23.7** |
| code_x_glue | 0.80 | 0.59 | **0.86** |

Table 9: Comparison on sample efficiency pre-trained after 50B tokens. We record selected tokens to achieve the performance of Full Data pre-trained with 50B tokens. The ablation interval is 0.5%.

| Setting | Full-Data | DSIR | QuRating-W | QuRating-A | D4 | DiSF (Ours) |
|---|---|---|---|---|---|---|
| TinyLlama-120M | 1.00 | 0.60 | 0.40 | 0.15 | 0.20 | 0.15 |
| TinyLlama-560M | 1.00 | 0.80 | 0.45 | 0.20 | 0.25 | 0.20 |
| TinyLlama-1.1B | 1.00 | 0.80 | 0.40 | 0.25 | 0.25 | 0.20 |

Table 10: Comparison on training efficiency pre-trained with 1.5% selection ratio. We record pre-trained tokens to achieve the performance of Full Data pre-trained with 50B tokens. - denotes the method can not reach that performance. The ablation interval is 1B.

| Setting | Full-Data | DSIR | QuRating-W | QuRating-A | D4 | DiSF (Ours) |
|---|---|---|---|---|---|---|
| TinyLlama-120M | 50B | - | - | 32B | 36B | 27B |
| TinyLlama-560M | 50B | - | - | 46B | 47B | 36B |
| TinyLlama-1.1B | 50B | - | - | 52B | 56B | 40B |

### A.3 BASELINES

#### A.3.1 DSIR.

DSIR (Xie et al., 2023b) treats Books and Wikipedia as high-quality targets for file selection, employing a hashed n-gram feature extractor to measure the similarity between the text features and the target distribution. In our experiments, following the selection procedures outlined in DSIR, we calculate importance scores using raw data (SlimPajama) and target data (Wikipedia and Books) in an n-gram feature space. The importance weights are then applied to resample a subset of the raw dataset. As for files in Wikipedia and Books domains, we proportionally integrate into the selected dataset.

#### A.3.2 QURATING.

QuRating (Wettig et al., 2024) queries GPT-3.5-turbo to train a judge model, that assess the specific quality of text samples, including four criteria: writing style, required expertise, facts & trivia, and educational value. In this paper, for Qurating-W, we select samples with the highest scores for writing style, while for QuRating-A, we proportionally select top-scoring samples across all four criteria.

#### A.3.3 DOREMI.

Doremi (Xie et al., 2023a), a recently proposed method that produces domain weights for pre-training on multiple text domains. In our experiments, we follow the domain weights calculation process of Doremi, using the domain weights from the initial data distribution as an initial reference to train a small reference model on the SlimPajama dataset. We then leverage this reference model to guide the training of a small proxy model, designed to generate domain weights. Finally, these domain weights are employed as the random selection ratio for text domains to construct selected dataset.

#### A.3.4 INGENIOUS

INGENIOUS (Renduchintala et al., 1991) extracts features using a model after a warm-up phase and employs Facility Location (Salhi, 1991a) to design a proxy function for feature importance, which measures the similarity between samples in the embedding space. Notably, as shown in Table 6, we observed that INGENIOUS, when using a warmed-up model for feature extraction (INGENIOUS-W), does not achieve satisfactory performance with our selected feature extractor, Contriever (INGENIOUS-C). Although INGENIOUS-C performs competitively compared to Random, DSIR, QuRating-W, and QuRating-A, our method consistently achieves the best performance across all settings.

#### A.3.5 D4.

The recent method D4 (Tirumala et al., 2023) notices the importance of diversified selection, involving SemDeDup (Abbas et al., 2024) and Prototypicality (Sorscher et al., 2022) to reduce file redundancy, but can not achieve satisfactory uniform representations as ours. In this paper, we sequentially applied the SemDeDup and Prototypicality methods to filter the data, controlling the filtering ratios of these two steps to be $R_{\text{dedup}} = 0.75$ and $R_{\text{proto}} = 0.02$, respectively.

Table 11: Model structure and training details of pre-training TinyLlama 120M.

| Parameter name | Value |
| --- | --- |
| Parameter number | 121,129,728 |
| Hidden size | 768 |
| Intermediate Hidden Size | 2048 |
| Context Len | 2048 |
| Heads | 12 |
| Layers | 12 |
| Vocab size | 32000 |
| Minimum learning rate | 4e-5 |
| Maximum learning rate | 4e-4 |
| Optimizer | AdamW |
| $\beta_1$ of optimizer | 0.9 |
| $\beta_2$ of optimizer | 0.95 |
| Warmup steps | 2000 |
| Batch size | 2M tokens |
| Weight decay | 0.1 |
| Activation function | SwiGLU |
| Gradient clipping threshold | 1.0 |
| Platform | 8 NVIDIA GeForce RTX 4090 GPUs |
| Training times on 10B tokens | about 0.2 days |
| Training times on 50B tokens | about 1 days |

Table 12: Model structure and training details of pre-training TinyLlama 560M.

| Parameter name | Value |
| --- | --- |
| Parameter number | 561,072,128 |
| Hidden size | 2048 |
| Intermediate Hidden Size | 2048 |
| Context Len | 2048 |
| Heads | 16 |
| Layers | 20 |
| Vocab size | 32000 |
| Minimum learning rate | 4e-5 |
| Maximum learning rate | 4e-4 |
| Optimizer | AdamW |
| $\beta_1$ of optimizer | 0.9 |
| $\beta_2$ of optimizer | 0.95 |
| Warmup steps | 2000 |
| Batch size | 2M tokens |
| Weight decay | 0.1 |
| Activation function | SwiGLU |
| Gradient clipping threshold | 1.0 |
| Platform | 8 NVIDIA GeForce RTX 4090 GPUs |
| Training times on 10B tokens | about 0.9 days |
| Training times on 50B tokens | about 4.5 days |

## A.4 MODEL AND TRAINING DETAILS

For better clarity and reproducibility, we provide the model structures and training details for pretraining TinyLlama with 120M, 560M, and 1.1B parameters, as shown in Tables 11, 12, and 13, respectively. These tables detail the configurations used in our experiments, facilitating easier

Table 13: Model structure and training details of pre-training TinyLlama 1.1B.

| Parameter name | Value |
| --- | --- |
| Parameter number | 1,100,048,384 |
| Hidden size | 2048 |
| Intermediate Hidden Size | 5632 |
| Context Len | 2048 |
| Heads | 32 |
| Layers | 22 |
| Vocab size | 32000 |
| Minimum learning rate | 4e-5 |
| Maximum learning rate | 4e-4 |
| Optimizer | AdamW |
| $\beta_1$ of optimizer | 0.9 |
| $\beta_2$ of optimizer | 0.95 |
| Warmup steps | 2000 |
| Batch size | 2M tokens |
| Weight decay | 0.1 |
| Activation function | SwiGLU |
| Gradient clipping threshold | 1.0 |
| Platform | 8 NVIDIA GeForce RTX 4090 GPUs |
| Training times on 10B tokens | about 1.7 days |
| Training times on 50B tokens | about 8.5 days |

replication of our results. Please note that our goal in this paper is not to optimize all hyper-parameters for the best LLM, but rather to compare selection methods under fair and reasonable conditions.

## A.5 MORE ANALYSIS

### A.5.1 PERFORMANCE ON LARGER DATASET AND MODEL SCALE

To evaluate the effectiveness of our selection method on larger models and datasets, we 1) pre-train Open-Llama 3B (Geng & Liu, 2023) on SlimPajama, and 2) TinyLlama-1.1B on both SlimPajama and StarcoderData (Li et al., 2023). StarcoderData is a dataset created for code generation, containing approximately 210M files (500GB of data). We assess the pre-trained models on common sense ability (7 tasks) and an additional code task, code_x_glue using harness evaluation. As shown in Table 7, we pre-train open-llama 3B with 10B training budget and 1.5% selection budget with SlimPajama. We compare our method with Random, DSIR, QuRating-A, and QuRating-W. DSIR and QuRating-W meet performance degradation due to dimensional collapse, while our method achieves the best performance on all tasks. As shown in Table 8, we pre-train TinyLlama-1.1B on both SlimPajama and StarcoderData with 1.5% selection budget and 10B training budget, compared to random selection and DSIR. Results of DSIR show performance degradation in code ability. We analyze that, DSIR tends to ignore code files in StarcoderData due to the selection critetion is based on WikiPedia, which means dimensional collapse happens.

### A.5.2 TRAINING EFFICIENCY COMPARED TO ALL BASELINES

To compare the computational and data efficiency of our approach with all baselines, we present additional results under the same settings as Figures 5 and 6, shown in Tables 9 and 10. The results demonstrate that, to achieve equivalent performance to Full Data with a 50B training budget, our method requires the fewest samples and pre-training tokens compared to the baselines, showing promising efficiency. Additionally, We also compare the cost of our method with all baselines in the process of selecting data. As reported in QuRating, annotating the data using the GPT API costs 520 NVIDIA H100 hours with additional ranking procedures of 3 hours. For DSIR, it takes more than 2 days using 48 CPUs in our platform. For DOREMI, training a 120M proxy model to provide weights for domains takes us approximately one week. In contrast, our method utilizes a public

feature extractor and selects samples in about 26 hours using one GPU and 48 CPUs. Combining these facts and our complexity analysis, we believe our method is practical among these methods for larger datasets.

### A.5.3 PROOF OF LOWER BOUND ON SUBMODULAR RATIO

We first recall our function, which is based on F-norm of the covariance matrix. The gain on A, and B with another sample e, can be formulated as following:

$$\Delta_A = e^{-\frac{1}{|A|+1}\sqrt{\sum_{i=1}^d \sum_{j=1}^d \left(\sum_{x\in A} x_i x_j + e_i e_j\right)^2}} - e^{-\frac{1}{|A|}\sqrt{\sum_{i=1}^d \sum_{j=1}^d \left(\sum_{x\in A} x_i x_j\right)^2}},$$

$$\Delta_B = e^{-\frac{1}{|B|+1}\sqrt{\sum_{i=1}^d \sum_{j=1}^d \left(\sum_{x\in B} x_i x_j + e_i e_j\right)^2}} - e^{-\frac{1}{|B|}\sqrt{\sum_{i=1}^d \sum_{j=1}^d \left(\sum_{x\in B} x_i x_j\right)^2}},$$

where $A \subset B \subset \Omega$, $e \in \Omega$, and $e \notin B$. We reformulate them as

$$\Delta_A = e^{-\frac{1}{|A|}\sqrt{\sum_i^d \sum_j^d (\sum_{x\in A} x_i x_j)^2}} \left(e^{\frac{1}{|A|}\sqrt{\sum_i^d \sum_j^d (\sum_{x\in A} x_i x_j)^2} - \frac{1}{|A+1|}\sqrt{\sum_i^d \sum_j^d (\sum_{x\in A} x_i x_j + e_i e_j)^2}} - 1\right),$$

$$\Delta_B = e^{-\frac{1}{|B|}\sqrt{\sum_i^d \sum_j^d (\sum_{x\in B} x_i x_j)^2}} \left(e^{\frac{1}{|B|}\sqrt{\sum_i^d \sum_j^d (\sum_{x\in B} x_i x_j)^2} - \frac{1}{|B+1|}\sqrt{\sum_i^d \sum_j^d (\sum_{x\in B} x_i x_j + e_i e_j)^2}} - 1\right).$$

We then define

$$\Delta(e|A) = \frac{1}{|A+1|}\sqrt{\sum_i^d \sum_j^d (\sum_{x\in A} x_i x_j + e_i e_j)^2} - \frac{1}{|A|}\sqrt{\sum_i^d \sum_j^d (\sum_{x\in A} x_i x_j)^2},$$

$$\Delta(e|B) = \frac{1}{|B+1|}\sqrt{\sum_i^d \sum_j^d (\sum_{x\in B} x_i x_j + e_i e_j)^2} - \frac{1}{|B|}\sqrt{\sum_i^d \sum_j^d (\sum_{x\in B} x_i x_j)^2}.$$

Therefore, the original equation will be:

$$\frac{\Delta_A}{\Delta_B} = e^{\frac{1}{|B|}\sqrt{\sum_i^d \sum_j^d (\sum_{x\in B} x_i x_j)^2} - \frac{1}{|A|}\sqrt{\sum_i^d \sum_j^d (\sum_{x\in A} x_i x_j)^2}} * \frac{e^{-\Delta(e|A)} - 1}{e^{-\Delta(e|B)} - 1}.$$

To provide a theoretical guarantee, we aim to establish a lower bound for the submodular ratio under any given set where the gain is positive, as Figure 4 generally suggest. Since the original formulation is challenging to analyze directly, we introduce two bounds on the function applied to the text files to derive a more concrete lower bound, as follows:

**Assumption 1** (Bounded Gain Difference). *Assume the gain difference between any two sets are bounded:*
$$|\Delta(e|A) - \Delta(e|B)| \leq \epsilon.$$

**Assumption 2** (Bounded Average Utility). *Assume average utility is $\mu$, which means $\forall U \in \Omega$, we have:*
$$\frac{1}{|U|}\sqrt{\sum_i^d \sum_j^d (\sum_{x\in U} x_i x_j)^2} \leq \mu.$$

With Assumption 1, we have:

$$\frac{\Delta_A}{\Delta_B} \geq e^{\frac{1}{|B|}\sqrt{\sum_i^d \sum_j^d (\sum_{x\in B} x_i x_j)^2} - \frac{1}{|A|}\sqrt{\sum_i^d \sum_j^d (\sum_{x\in A} x_i x_j)^2}} * \frac{e^{-\Delta(e|B)-\epsilon} - 1}{e^{-\Delta(e|B)} - 1}.$$

With Assumption 2, we have:

$$\frac{1}{|B|}\sqrt{\sum_i^d \sum_j^d (\sum_{x\in B} x_i x_j)^2} - \frac{1}{|A|}\sqrt{\sum_i^d \sum_j^d (\sum_{x\in A} x_i x_j)^2} \geq -2\mu,$$

which means $\frac{\Delta_A}{\Delta_B} \geq e^{-2\mu} * \frac{e^{-\Delta(e|B)-\epsilon}-1}{e^{-\Delta(e|B)}-1}$. We reformulate this as

$$\frac{\Delta_A}{\Delta_B} \geq e^{-2\mu} * (e^{-\epsilon} + \frac{e^{-\epsilon}-1}{e^{-\Delta(e|B)}-1}).$$

From Assumption 2, we have $\frac{1}{|B|}\sqrt{\sum_i^d \sum_j^d (\sum_{x \in B} x_i x_j)^2} \leq \mu$. Therefore $e^{-\Delta(e|B)} \leq e^{2\mu}$, and we have:

$$\frac{\Delta_A}{\Delta_B} \geq e^{-2\mu} * (e^{-\epsilon} + \frac{e^{-\epsilon}-1}{e^{2\mu}-1}).$$

Finally, we have a lower bound as $e^{-2\mu}\frac{e^{2\mu-\epsilon}-1}{e^{2\mu}-1}$, which means given assumptions 1, 2, and positive gain, our function is at least $e^{-2\mu}\frac{e^{2\mu-\epsilon}-1}{e^{2\mu}-1}$-weakly submodular function.

### A.5.4 COMPLEXITY ANALYSIS

For a detailed analysis of time complexity, we divide it into two parts: 1) computational complexity shown in Algorithm 1. In each batch, we initialize $U_i$ with a randomly selected sample and remove it from the batch. Then, we iteratively apply $(\lfloor\frac{b|\mathcal{S}|}{|\mathbb{D}|}\rfloor - 1)$ times the Argmax command on the batch of data with our proxy fuction. Denote the computational cost of our proxy function with k text samples as $F_{|U|=k}(U) = OF_k)$, the computation cost will be:

$$O(1+...+\frac{|\mathbb{D}|}{b})\sum_{k=1}^{\frac{b|\mathcal{S}|}{|\mathbb{D}|}}(b-k)(F_{k+1}) \leq O(\frac{|\mathbb{D}|^2}{b^2})\frac{b|\mathcal{S}|}{|\mathbb{D}|}bF_{k+1} = O(|\mathbb{D}|\mathcal{S}F_{\frac{b|\mathcal{S}|}{|\mathbb{D}|}+1}),$$

where b is the batch scale, $|\mathbb{D}|$ is the total data scale, $|\mathcal{S}|$ is the selection budget. 2) The complexity of proxy function $O(F_k)$. Given text features $z$ and their feature dimension d, Frobenius norm and $z \cdot z^T$ are both $O(d^2)$. Since our proxy function calculates k times the $z \cdot z^T$, $O(F_k) = kd^2$. Finally, the complexity of our DiSF will be:

$$O(DiSF) \leq O(|\mathbb{D}|\mathcal{S}F_{\frac{b|\mathcal{S}|}{|\mathbb{D}|}+1}) = O(|S|^2 bd^2)$$

All terms in the time complexity are at most quadratic and independent of the overall dataset size, which we consider acceptable for applying to larger datasets. The space complexity largely depends on the stored features of all text files: $O(|\mathbb{D}|d^2)$.

Table 14: Pre-trained performance compared to DiSF-LD.

| Setting | Random | DSIR | D4 | QuRating-W | QuRating-A | DiSF-LD | DiSF (Ours) |
|---|---|---|---|---|---|---|---|
| TinyLlama120M | 38.9 | 37.8 | 39.7 | 38.9 | 40.0 | 39.5 | 40.6 |

### A.5.5 COMPARISON TO SUBMODULAR FUNCTIONS

In this section, we compare our proxy function with two strictly submodular functions: facility location and log-determinant. For the facility location function, we compare with INGENIOUS. Additionally, we introduce a variant defined as $F_{LD}$=LogDet(I+C) (DiSF-LD), where $I$ is the identity matrix and $C$ is covariance matrix. We evaluate this variant alongside our original proxy function on TinyLlama-120M, using a 1.5% selection ratio and a 50B training budget. As shown in Tables 2, and 14, the results demonstrate that both DiSF-LD and INGENIOUS help mitigate dimensional collapse and improve the performance of pre-trained LLMs. However, neither method achieves the same level of performance as our original DiSF. This is due to their inability to directly optimize the uniformity of feature dimensions, leading to a trade-off between strict submodularity and the specific goal of optimizing dimensional uniformity.

### A.5.6 OBSERVATION OF DIMENSIONAL COLLAPSE ON STARCODER

As shown in Table 15, we present the selection results of DSIR on Starcoderdata, as well as on both Starcoderdata and SlimPajama, across two domains: Wikipedia and Python. The results demonstrate

Table 15: File selection of DSIR on Starcoder. We denote selected file ratios of DSIR based on Wikipedia domain under both Starcoderdata and SlimPajama as DSIR-W-SS , on Wikipedia domain under Starcoderdata as DSIR-W-S and Python domain under Starcoderdata as DSIR-P-S.

| Method | go | java | javascript | php | python | ruby | slimpajama |
|---|---|---|---|---|---|---|---|
| Original | 0.59% | 2.52% | 2.45% | 1.97% | 1.61% | 0.43% | 74.07% |
| DSIR-W-SS | 0.00% | 0.77% | 0.54% | 0.00% | 0.5% | 0.00% | 94.0% |
| DSIR-W-S | 0.03% | 12.37% | 17.15% | 0.12% | 7.58% | 0.00% | - |
| DSIR-P-S | 0.00% | 25.04% | 4.36% | 2.20% | 50.38% | 0.00% | - |

that DSIR, when applied to a single domain, tends to select similar files, indicating dimensional collapse. Notably, during our analysis of the scores output by QuRater, we found that the writing style scores for most files in Starcoderdata are negative. As a result, these files are unlikely to be selected when combined with SlimPajama, further highlighting the issue of dimensional collapse.

## A.6 PROOF OF LEMMA 1

**Lemma 2.** *Assuming a covariance matrix $M \in \mathbf{R}^{d \times d}$ computed from the feature of each sample with the standard normalization, and its eigenvalues $\{\lambda_1, \lambda_2, ..., \lambda_d\}$, we will have the following equality that satisfied*

$$\sum_{i=1}^{d} (\lambda_i - \frac{1}{d} \sum_{j=1}^{d} \lambda_j)^2 = ||M||_F^2 - d.$$

*Proof.* Let $M \in \mathbb{R}^{d \times d}$ be a covariance matrix computed from features that have been standardized—that is, the data has been centered (zero mean) and scaled to have unit variance along each dimension. This standard normalization implies that the trace of $M$ equals $d$:

$$\mathrm{Tr}(M) = \sum_{i=1}^{d} \lambda_i = d.$$

This means the average eigenvalue is:

$$\bar{\lambda} = \frac{1}{d} \sum_{i=1}^{d} \lambda_i = 1.$$

**Left-Hand Side (LHS) Calculation:**

The left-hand side involves the sum of squared deviations of the eigenvalues from their mean:

$$\sum_{i=1}^{d} \left( \lambda_i - \frac{1}{d} \sum_{j=1}^{d} \lambda_j \right)^2 = \sum_{i=1}^{d} \left( \lambda_i - \bar{\lambda} \right)^2 = \sum_{i=1}^{d} (\lambda_i - 1)^2.$$

Expanding each term on the right, we get the following equation:

$$\sum_{i=1}^{d} (\lambda_i - 1)^2 = \sum_{i=1}^{d} \left( \lambda_i^2 - 2\lambda_i + 1 \right)$$

$$= \sum_{i=1}^{d} \lambda_i^2 - 2 \sum_{i=1}^{d} \lambda_i + \sum_{i=1}^{d} 1.$$

Simplifying with $\sum_{i=1}^{d} \lambda_i = d$ and $\sum_{i=1}^{d} 1 = d$, we have:

$$\sum_{i=1}^{d} \lambda_i^2 - 2d + d = \sum_{i=1}^{d} \lambda_i^2 - d.$$

**Right-Hand Side (RHS) Calculation:**

First, the Frobenius norm of a matrix $M$ is defined as

$$\|M\|_F^2 = \sum_{i=1}^{d} \sum_{j=1}^{d} M_{ij}^2 = \text{Tr}(M^\top M).$$

Since $M$ is symmetric ($M = M^\top$), this simplifies to:

$$\|M\|_F^2 = \text{Tr}(M^2).$$

Using the spectral theorem, we have

$$M = U\Lambda U^\top,$$

where $U$ is an orthogonal matrix whose columns are the eigenvectors of $M$, and $\Lambda = \text{diag}(\lambda_1, \lambda_2, \ldots, \lambda_d)$ contains the eigenvalues. Since $U^\top U = I$, we have the following equation:

$$M^2 = (U\Lambda U^\top)(U\Lambda U^\top) = U\Lambda U^\top U\Lambda U^\top = U\Lambda^2 U^\top,$$

Therefore, the trace of $M^2$ is

$$\text{Tr}(M^2) = \text{Tr}(U\Lambda^2 U^\top) = \text{Tr}(\Lambda^2 U^\top U) = \text{Tr}(\Lambda^2) = \sum_{i=1}^{d} \lambda_i^2.$$

Thus, the right-hand side is

$$\|M\|_F^2 - d = \left( \sum_{i=1}^{d} \lambda_i^2 \right) - d.$$

**Conclusion:**

Comparing both sides:

$$\sum_{i=1}^{d} \left( \lambda_i - \frac{1}{d} \sum_{j=1}^{d} \lambda_j \right)^2 = \sum_{i=1}^{d} \lambda_i^2 - d = \|M\|_F^2 - d.$$

This completes the proof.

$\square$

