# OpenReview forum: "Combatting Dimensional Collapse in LLM Pre-Training Data via Submodular File Selection"
_ICLR.cc/2025/Conference — ICLR 2025 Oral_

### Official Review · Reviewer_8JRU · 2024-10-31

**Soundness:** 2
**Presentation:** 2
**Contribution:** 2
**Rating:** 8
**Confidence:** 2

**Summary:**

This paper proposes a novel file selection method for LLM pretraining to avoid the dimensional collapse issue. The method can achieve greater uniformity across feature dimensions, while the other methods usually get a long narrow feature space, limiting their performance on all the tasks.

**Strengths:**

1. This paper observes the drawbacks of the other file selection methods for LLM pretraining, which usually obtain quite narrow feature space, and are hard to obtain good results for all the tasks.

2. The proposed method is interesting, which converts the uniformity of the eigenvalues of the covariance matrix into minimizing the F norm of the covariance matrix, then uses a submodular optimization to solve it.

3. Experimental results on 7 tasks show the proposed method achieves a higher average perfomance than the other baselines, and is quite data efficient compared to full data pretraining.

**Weaknesses:**

1. The observation in figure 2 is evaluated on the SlimPajama. Do these methods have the same phenomenan on the other pretraining corpus.

2. In the efficiency analysis, comparing with full data pretraining is not enough, the comparison with the other baselines is not given.

**Questions:**

What does (1-e^-1) approximation mean in the abstract.

---

> ### Author Response · Authors · 2024-11-20
> **Rebuttal to 8JRU (1)**
>
> **Thanks for the constructive feedback provided by the Reviewer 8JRU. We sincerely appreciate the time and effort you dedicated to evaluating our work. Below, we provide detailed responses to the weaknesses and questions, which we denote as W and Q. Besides, we also added the following discussions into revised submission.**
>
> >**W1**: The observation in figure 2 is evaluated on the SlimPajama. Do these methods have the same phenomenan on the other pretraining corpus.
>
> **Reply to W1:**
> **Thanks for the valuable comments! We observe the similar phenomenon (dimensional collapse) with additional text corpora, starcoderdata** (210M files, approximately 500GB) [1], which is specialized for code generation. In the following, we verify the phenomenan from three views:
> 1) **Selecting similar files on other pretraining corpora.** As shown in following Table T1, we present the selection results of DSIR on Starcoderdata and both Starcoderdata and Slimpajama involving two domains: Wikipedia and Python. All results demonstrate that DSIR, when operating on a single domain, tends to select similar files, which indicates dimensional collapse. Experiments with QuRating are still ongoing, as it requires running its judger model (QuRater-1.3B) to evaluate all files, and we expect to complete it in the coming days. Notably, during our analysis of the scores output by QuRater, we found that the writing style scores for most files in Starcoderdata are negative. This makes these files unlikely to be selected when combined with SlimPajama, further underscoring the issue of dimensional collapse.
> 2) **Larger dominance score on other pretraining corpora.** As shown in following Table T2, we present the dominance score$\frac{\sum_{i=1}^k\lambda_i}{\sum_{j=1}^d\lambda_j}$ of feature covariance matrix for files selected by DSIR respectively based on the Wikipedia domain and the Python domain, compared to random selection and our DiSF. Results show that regardless of the domain DSIR is based on, the dominance score of its selected files is higher than that of random selection, let alone our method. This indicates a severe issue of dimensional collapse in DSIR's selected files.
> 3) **Performance degradation on other pretraining corpora.** As shown in following Table T3, we compare our method with DSIR (based on the Python domain), random selection, and Full Data pre-training. The model is evaluated on the 'code_x_glue' task [2] using the Harness framework on TinyLlama-120M. This task consists of six sub-tasks: Go, Java, JavaScript, PHP, Python, and Ruby. The results show that while DSIR performs better on Python and Java, it achieves poor average performance across the sub-tasks, highlighting dimensional collapse. In contrast, our method achieves the best average performance among all compared baselines.
>
>
> **Table T1**: Selected file ratios of DSIR based on Wikipedia domain under both Starcoderdata and SlimPajama (DSIR-W-SS), on Wikipedia domain under Starcoderdata (DSIR-W-S) and Python domain under Starcoderdata (DSIR-P-S).
>
> | Method    | go    | java   | javascript | php   | python | ruby  | slimpajama | donimance score |
> | --------- | ----- | ------ | ---------- | ----- | ------ | ----- | ---------- | --------------- |
> | Original  | 0.59% | 2.52%  | 2.45%      | 1.97% | 1.61%  | 0.43% | 74.07%     | 0.5715          |
> | DSIR-W-SS | 0.00% | 0.77%  | 0.54%      | 0.00% | 0.5%   | 0.00% | 94.0%      | 0.6947          |
> | DSIR-W-S  | 0.03% | 12.37% | 17.15%     | 0.12% | 7.58%  | 0.00% | -          | 0.5912          |
> | DSIR-P-S  | 0.00% | 25.04% | 4.36%      | 2.20% | 50.38% | 0.00% | -          | 0.6833          |
>
>
>  **Table T2** Dominance score compared on starcoderdata.
> | Setting| Random| DSIR-Wikipedia|DSIR-Python|DiSF|
> | - | - | - | - | - |
> | k=20 |  0.171 | 0.259 | 0.302 | 0.127 |
> | k=40 | 0.295 |0.360 | 0.443 | 0.250 |
> | k=60 |  0.374 | 0.448 | 0.536 | 0.334 |
> | k=80 | 0.459 | 0.526 | 0.611 | 0.395 |
> | k=100 |0.527  | 0.591 | 0.683 | 0.461 |
>
> **Table T3**: Code ability on six sub-tasks with 1.5% selection and 50B training budget on TinyLlama120M.
> | Method     | go | java | javascript | php| python | ruby |
> | - | - | - | - | - | - | - |
> | Full Data    | 0.89 | 0.83 | 0.84       | 0.75 | 0.92   | 0.76 |
> | Random       | 0.74 | 0.71 | 0.77       | 0.62 | 0.85   | 0.63 |
> | DSIR-P       | 0.61 | 0.77 | 0.72       | 0.56 | 0.88   | 0.54 |
> | DiSF (Ours)  | 0.86 | 0.77 | 0.79       | 0.70 | 0.89   | 0.68 |
>
> [1] Starcoder: may the source be with you!
>
> [2] CodeXGLUE: A Machine Learning Benchmark Dataset for Code Understanding and Generation

---

> > ### Author Response · Authors · 2024-11-20
> > **Rebuttal to 8JRU (2)**
> >
> > >**W2**: In the efficiency analysis, comparing with full data pretraining is not enough, the comparison with the other baselines is not given.
> >
> > **Reply to W2:**
> > **We appreciate the feedback to improve our submission and have included the following results in the revised version.** To compare the computational and data efficiency of our approach with all baselines, we present additional results under the same settings as Figures 5 and 6, shown in following Tables T4 and T5. The results demonstrate that, to achieve equivalent performance to Full Data with a 50B training budget, our method requires the fewest samples and pre-training tokens compared to the baselines, showing promising efficiency.
> >
> >
> > **Table T4**: Data efficiency. Required sample ratios to meet the performance of Full-Data with 50B training budget. Grid search ratio is 0.05.
> > | Setting     |  DSIR  | QuRating-W | QuRating-A| D4 | DiSF (Ours) |
> > | ----------- | ----------| ------| -----------|-----------|-----------|
> > | TinyLlama120M |  0.6 | 0.4  | 0.15 | 0.2 | 0.15 |
> > | TinyLlama560M |  0.8 | 0.45 |  0.2| 0.25 | 0.2 |
> > | TinyLlama1.1B | 0.8 | 0.4 | 0.25 | 0.25 | 0.2 |
> >
> >
> > **Table T5**: Computational efficiency. Required pre-trained tokens to meet the performance of Full-Data with 50B training budget budget and 1.5% selection budget. - denotes the method can not reach that performance.
> > | Setting     |  DSIR  | QuRating-W | QuRating-A| D4 | DiSF (Ours) |
> > | ----------- | ----------| ------| -----------|-----------|-----------|
> > | TinyLlama120M |  - | -  | 32B | 36B | 27B |
> > | TinyLlama560M |  - | - |  46B| 47B | 36B |
> > | TinyLlama1.1B | - | - | 52B | 56B | 40B |
> >
> >
> >
> > >**Q1**: What does (1-e^-1) approximation mean in the abstract.
> >
> >
> > **Reply to Q1:**
> >
> > Given the set, set function, selection algorithm, and selection budget, the $(1 - e^{-1})$ approximation [3] to the optimal solution refers to $\frac{\text{approximated function value via the algorithm under the selection budget}}{\text{optimal function value under the selection budget}}\geq 1-\frac{1}{e}$. **We have revised the submission and provided a more detailed explanation to make it clearer for readers.**
> >
> > [3] An analysis of approximations for maximizing submodular set functions

---

> > > ### Author Response · Authors · 2024-11-23
> > > **Invitation to rolling discussion for the possible remaining concerns**
> > >
> > > Dear Reviewer 8JRU,
> > >
> > > **We truly appreciate the effort and time you have devoted to providing constructive reviews, as well as your positive evaluation of our submission.** We have now provided more clarifications, explanations, and experiments to address your concerns and followed your advice to improve our paper. Here is a summary for you:
> > >
> > > * We explain and refine some desciptions.
> > > * We provide the support of dimensional collpase on another dataset: Starcoderdata.
> > > * We provide additional efficiency analysis compared with all baselines.
> > >
> > > **Would you mind checking our responses and confirming if you have any additional questions?** We truly appreciate this opportunity to improve our work and shall be most grateful for any feedback you could give to us.

---

> > > > ### Comment · Reviewer_8JRU · 2024-11-27
> > > > **Thanks for your reply.**
> > > >
> > > > The authors have addressed my concerns, and I decide to increase my rating!

---

> > > > > ### Author Response · Authors · 2024-11-27
> > > > > **Thanks for the reply and the increased score!**
> > > > >
> > > > > We thank for the endorsement of the reviewer and the increased score! We will keep improving our paper, following your valuable advice.

---

### Official Review · Reviewer_7MY9 · 2024-11-01

**Soundness:** 3
**Presentation:** 4
**Contribution:** 4
**Rating:** 8
**Confidence:** 4

**Summary:**

The authors propose a diversified submodular file selection algorithm (DiSF) to select sample set with more diverse data. It can alleviate dimensional collapse problem in feature space and enable LLM model to have generic performance.

**Strengths:**

The paper addresses an important problem in LLM pretraining and proposes a simple yet effective method. The writing is clear and the narrative is easy to follow, facilitating understanding of complex concepts.

**Weaknesses:**

No major weakness.

**Questions:**

1. It is recommended to provide the sample size and performance analysis experiments under the full data case and data selection case in different datasets.
2. It is suggested to provide time complexity and space complexity metrics on Table 2. The experiment in Section 'Selection scale' is not enough to illustrate the computational efficiency of the proposed method.
3. The performance improvement in Table 2 is limited, most of which are within ±0.5 (within the error range), and reasonable analysis is needed to illustrate the effectiveness of the method.

---

> ### Author Response · Authors · 2024-11-20
> **Rebuttal to 7MY9 (1)**
>
> **Thanks for the valuable comments provided by the Reviewer 7MY9. We sincerely appreciate the time and effort you dedicated to evaluating our work. Below, we provide detailed responses to the questions, which we denote as Q. Besides, we also added the following discussions into revised submission. The corresponding reference is listed at last response.**
>
> >**Q1**: It is recommended to provide the sample size and performance analysis experiments under the full data case and data selection case in different datasets.
>
> **Reply to Q1:**
> **We conduct sample size and performance analysis on another dataset, Starcoderdata** [1] (210M files, approximately 500GB), which is specialized for code generation. The model is evaluated on the 'code_x_glue' [2] task using the Harness framework on TinyLlama-120M. This task comprises six sub-tasks, including Go, Java, JavaScript, PHP, Python, and Ruby. We present results compared to Full Data, Random Selection, and DSIR. **Comparison with QuRating is ongoing, as it requires running its judger model (QuRater-1.3B) to evaluate all files, and we anticipate completing it in the coming days.** Below, we provide detailed analysis on: 1) experiments related to dimensional collapse, 2) performance comparisons on sub-tasks, and 3) performance comparisons across different sample sizes.
>
> 1) **Dimensional collapse is observed in another dataset with DSIR and QuRating-W.** As shown in following Table T1, we present the selected results for SlimPajama and Starcoderdata using DSIR across two domains: Wikipedia and Python. All results demonstrate that DSIR, when operating on a single domain, tends to select similar files, indicating dimensional collapse. Additionally, while processing the scores output by QuRater, we found that the writing style scores for most files in Starcoderdata are negative. This makes them unlikely to be selected when combined with SlimPajama, further highlighting the issue of dimensional collapse.
> 2) **Best averaged performance on another dataset.** As shown in following Table T2, we compare our method with DSIR (based on the Python domain), random selection, and Full Data pre-training on Starcoderdata. The results show that DSIR performs better on Python and Java but achieves poor averaged performance across subtasks. In contrast, our method achieves the best averaged performance among all selection baselines.
> 3) **Scalability with Sample Size on another dataset.** As shown in following Table T3, we compare our method with Full Data, Random selection, and DSIR (based on the Python domain) across different sample sizes. The results demonstrate our method consistently achieves the best performance under different sample sizes, compared to selection methods. Besides, under a larger sample scale (exceed 3%), our method outperforms Full-Data.
>
> **Table T1**: Selected file ratios of DSIR based on Wikipedia domain under both Starcoderdata and SlimPajama (DSIR-W-SS), on Wikipedia domain under Starcoderdata (DSIR-W-S) and Python domain under Starcoderdata (DSIR-P-S).
>
> | Method | go | java | javascript | php| python | ruby |slimpajama|donimance score|
> | - | - | - | - | - | - | - | - | - |
> | Original|0.59%| 2.52%|2.45%|1.97%| 1.61%  | 0.43%  | 74.07% | 0.5715 |
> | DSIR-W-SS|0.00%|0.77%|0.54% | 0.00% | 0.5%  | 0.00%  | 94.0% | 0.6947 |
> | DSIR-W-S | 0.03%| 12.37%|17.15%|0.12%|7.58% | 0.00%  | - | 0.5912 |
> | DSIR-P-S|0.00% |25.04%|4.36%|2.20%|50.38%| 0.00%  | - | 0.6833 |
>
> **Table T2**: Programming ability on six sub-tasks with 1.5% selection and 50B training budget on TinyLlama120M.
> | Method     | go | java | javascript | php| python | ruby |avg|
> | - | - | - | - | - | - | - | - |
> | Full Data    | 0.89 | 0.83 | 0.84| 0.76|0.92| 0.76 | 0.83 |
> | Random |0.74|0.71|0.77|0.63|0.85| 0.63 |0.72 |
> | DSIR-P|0.61| 0.77 | 0.72| 0.56 | 0.89 | 0.54 |0.68 |
> | DiSF (Ours) |0.86|0.77| 0.79| 0.71 | 0.89   | 0.68 | 0.78 |
>
> **Table T3**: Problem solving and programming ability (averaged performance on six sub-tasks) on TinyLlama-120M pre-trained on starcoderdata with different sample size and 50B training budget.
> | Sample Size | Task        | Full Data | Random | DSIR  | DiSF (Ours) |
> | ----------- | ----------- | --------- | ------ | ----- | ----------- |
> | 0.5%        | bbh  | 14.97     | 10.40  | 10.04 | 12.07       |
> |             | mmlu  | 23.52     | 20.87  | 20.33 | 22.02       |
> |             | code_x_glue | 0.85      | 0.68   | 0.65  | 0.74        |
> | 1.5%        | bbh | 14.97| 12.51  | 12.54 | 14.53  |
> |             | mmlu | 23.52 | 21.95  | 21.27 | 22.75 |
> |             | code_x_glue | 0.85      | 0.73   | 0.69  | 0.79  |
> | 3%          | bbh | 14.97 | 13.35  | 12.79 | **15.39**  |
> |             | mmlu  | 23.52 | 22.88  | 22.33 | **23.67**  |
> |             | code_x_glue | 0.85      | 0.82   | 0.76  | **0.89**        |
>
> [1] Starcoder: may the source be with you!
>
> [2] CodeXGLUE: A Machine Learning Benchmark Dataset for Code Understanding and Generation

---

> > ### Author Response · Authors · 2024-11-20
> > **Rebuttal to 7MY9 (2)**
> >
> > >**Q2**: It is suggested to provide time complexity and space complexity metrics on Table 2. The experiment in Section 'Selection scale' is not enough to illustrate the computational efficiency of the proposed method.
> >
> > We thanks for the constructive advide! We have made following discussion into the revised submission to better illustrate the efficiency of our method. **The time complexity of our method is at most $O(|\mathcal{S}|^2bd^2)$, and the required space depends on the stored features of all text files: $O(|\mathbb{D}|d^2)$**, where $\mathcal{S}$ is selection budget, $\mathbb{D}$ is the text corpus, b is batch scale, and $d$ is the dimension of text features. For other methods, their space and time complexities are difficult to analyze theoretically. For example, QuRating utilizes the GPT-3.5 API to train a 1.3B rater, DOREMI trains an additional 120M proxy model, and D4 employs techniques such as SemDeDup and Prototypicality, making their complexities hard to determine. To better illustrate the computational efficiency of our method in both the selection and pre-training stages, we provide additional details as follows:
> > 1) According to QuRating, annotating the data using the GPT API requires 520 NVIDIA H100 GPU hours. For DSIR, the process takes more than two days using 48 CPUs on our platform. For DOREMI, training a 120M proxy model to generate domain weights takes approximately one week on our platform. In contrast, our method leverages a public feature extractor and completes sample selection in about 26 hours using a single GPU and 48 CPUs.
> > 2) To better illustrate the computational efficiency during LLM pre-training, we additionally record the training acceleration of all baselines in following Table T3. The results demonstrate that our method achieves superior training efficiency compared to all other methods.
> > 3) Here we provide a detaild analysis about time complexity. We divide it into two parts: 1) Computational complexity shown in Algorithm 1 of the submission. In each batch, we initialize
> >  $\mathrm{U}$ with a randomly selected sample and remove it from the batch. Then, we iteratively apply $(\lfloor\frac{b|\mathcal{S}|}{|\mathbb{D}|}\rfloor-1)$ times the Argmax command on the batch of data with our proxy fuction. Denote the computational cost of our proxy function with k text samples as $F_{|U|=k}(U)=O(F_k)$, the computation cost will be:$$O(1+...+\frac{|\mathbb{D}|}{b})\sum_{k=1}^{\frac{b|\mathcal{S}|}{|\mathbb{D}|}}(b-k)(F_{k+1}) \leq O(\frac{|\mathbb{D}|^2}{b^2})\frac{b|\mathcal{S}|}{|\mathbb{D}|}b F_{k+1} =O(|\mathbb{D}|\mathcal{S}F_{\frac{b|\mathcal{S}|}{|\mathbb{D}|}+1}),$$ where b is the batch scale, $|\mathbb{D}|$ is the total data scale, $|\mathcal{S}|$ is the selection budget. 2) The complexity of proxy function $O(F_k)$. Given text features $z$ and their feature dimension d, Frobenius norm and $z\cdot z^T$ are both $O(d^2)$. Since our proxy function calculates k times the $z\cdot z^T$, therefore $O(F_k)=kd^2$. Finally, the complexity of our DiSF will be: $$O(DiSF)\leq O(|\mathbb{D}|\mathcal{S}F_{\frac{b|\mathcal{S}|}{|\mathbb{D}|}+1}) =O(|S|^2bd^2)$$
> >
> > **Table T3**: Training efficiency. Required pre-trained tokens to meet the performance of Full-Data with 50B training budget budget and 1.5% selection budget. - denotes the method can not reach that performance.
> > | Setting     |  DSIR  | QuRating-W | QuRating-A| D4 | DiSF (Ours) |
> > | ----------- | ----------| ------| -----------|-----------|-----------|
> > | TinyLlama120M |  - | -  | 32B | 36B | 27B |
> > | TinyLlama560M |  - | - |  46B| 47B | 36B |
> > | TinyLlama1.1B | - | - | 52B | 56B | 40B |

---

> > > ### Author Response · Authors · 2024-11-20
> > > **Rebuttal to 7MY9 (3)**
> > >
> > > >**Q3**: The performance improvement in Table 2 is limited, most of which are within ±0.5 (within the error range), and reasonable analysis is needed to illustrate the effectiveness of the method.
> > >
> > > **Reply to Q3:**
> > > **Thanks for the constructive advice for better improving our submission.** Here we provided more analysis to illustrate the effectiveness of our method. All following discussions are added to our revised submission.
> > > 1) **Our method achieves much smaller dominance score.** As shown in Figure 3 and Figure 4 of the submission, compared to all methods, our method achieve much smaller dominance score and quiet larger proxy value, indicating a better uniformaty of feature dimensions among selected files.
> > > 2) **Our method achieves promising training and sample efficiency.** As shown in Figure 5, and Figure 6 of original submission, our method achieves promising training and sample efficiency compared to Full Data pre-training. Besides, in the response to Q2 and Table T3, our method show superior training efficiency compared to all baselines.
> > > 3) **Our method achieves consistently better performance on larger model and more dataset.** To better illustrated the effectiveness of our method, we verify DiSF on more datasets (in response to Q1) and larger scale of LLM (OpenLlama-3B [3]) in the following Table T4. Both results demonstrate the consistently better performance of of DiSF compared to baselines.
> > >
> > > **Notably, it is not easy to improve the performance on those chanllenging tasks.** According to the **official TinyLlama report**, a 1.1B model pre-trained on 100B to 500B tokens across all data achieved an average performance improvement from 46.11 to 48.28 on seven common-sense tasks—a 2.1% gain. In our experiments, pre-training the same model with 10B to 50B tokens on 10B data, our method achieves a 3% improvement (from 42.2 to 45.2), while DSIR achieves a 1.7% improvement.
> > >
> > > **Table T4**: Comparison on openllama 3B with 1.5% selecting ratio and 10 B pre-training budget on SlimPajama.
> > > | Setting       | Random | DSIR | QuRating-W | QuRating-A | DiSF (Ours) |
> > > | ------------- | ---------------- | ---- | ---------- | ---------- | ----------- |
> > > | arc_challenge | 20.1   | 22.3  | **24.2**       | 23.3        | 22.1        |
> > > | arc_easy      | 47.1   | 46.6  | 49.2       | **52.1**        | 49.1        |
> > > | boolq         | 60.6   | 59.5  | 60.1       | 61.0        | **61.8**        |
> > > | hellaswag     | 33.9   | 29.7  | 30.7       | 33.2        | **34.4**        |
> > > | openbookqa    | 19.8   | 19.6  | 20.0       | 20.4        | **20.6**        |
> > > | piqa          | 67.7   | 65.5  | 62.9       | 63.0        | **69.9**        |
> > > | winogrande    | 52.2   | 49.8  | 50.9       | 52.3        | **52.8**        |
> > > | **Avg**       | 43.1   | 41.9  | 42.6       | 43.6        | **44.4**        |
> > >
> > > [3] Openllama: An open reproduction of llama

---

> > > > ### Author Response · Authors · 2024-11-23
> > > > **Invitation to rolling discussion for the possible remaining concerns**
> > > >
> > > > Dear Reviewer 7MY9,
> > > >
> > > > **We truly appreciate the effort and time you have devoted to providing constructive reviews, as well as your positive evaluation of our submission.** We have now provided more clarifications, explanations, and experiments to address your concerns and followed your advice to improve our paper. Here is a summary for you:
> > > >
> > > > * We discussed the time complexity and space complexity of our algorithm.
> > > > * We verify our method on more datasets.
> > > > * We provide the sample size and performance analysis experiments in a different dataset.
> > > > * We provide more analysis to illustrate the effectiveness of our method.
> > > >
> > > >
> > > > **Would you mind checking our responses and confirming if you have any additional questions?** We truly appreciate this opportunity to improve our work and shall be most grateful for any feedback you could give to us.

---

> > > > > ### Comment · Reviewer_7MY9 · 2024-12-02
> > > > >
> > > > > Thanks to you for the reply, I am happy to maintain this score.

---

### Official Review · Reviewer_H11o · 2024-11-03

**Soundness:** 3
**Presentation:** 3
**Contribution:** 2
**Rating:** 8
**Confidence:** 4

**Summary:**

The paper introduces the DiSF (Diversified Submodular File Selection) algorithm to combat dimensional collapse in Large-Language Model (LLM) pretraining. The DiSF algorithm is designed as a subset selection task with a goal of selecting a diverse subset from an unlabeled training corpus and is achieved by maximizing a submodular information function over the complete unlabeled set with sample interactions modeled through a normalized covariance matrix. The paper demonstrates upto 1.5x training efficiency and upto 5x data efficiency on several benchmark datasets.

**Strengths:**

The strengths of this paper can be summarized as below -

1. The paper address the task of data-efficient LLM pretraining while boosting performance on several benchmark tasks by minimizing the impact of dimensionality collapse which is a critical challenge in this domain.

2. The authors clearly demonstrate the impact of dimensionality collapse in existing approaches targeting data-efficient LLM training and establish the requirement for decorrelated text samples during the training process to minimize its effect.

3. The DiSF algorithm models this requirement as a diversity driven selection problem and achieves this my training the underlying LLM on a subset of data selected by maximizing a submodular function $C(U, M)$ over the complete unlabeled set $\mathbb{D}$. This proposed methodology is strongly supported by experimental evidence on multiple benchmark datasets.

**Weaknesses:**

Although the results demonstrated in the paper are impressive the following weaknesses/ questions need to be addressed.

1. Although the subset selection mechanism in DiSF depends on submodular maximization of $F_{M}^{DiSF}(U)$ but the proof of submodularity of $F_{M}^{DiSF}(U)$ has not been provided. The paper presents only an empirical proof of monotonicity through Figure 4 but that is not deemed sufficient (Fujishige, 2005) to prove a function to be submodular. The authors must provide proof of $F_{M}^{DiSF}(U)$ demonstrating the diminishing marginal returns property described in lines 232- 234.

2. Diversity driven subset selection for data-efficient LLM pretraining is not novel to the current paper. Prior work like INGENIOUS (Renduchintala et al., 2023) have demonstrated significant improvements in model performance while being data-efficient. The authors should compare against such methods to study the impact of dimensionality collapse and the diversity selection algorithm.

3. Submodular Functions like Log-Determinant, Facility-Location etc. have been shown to model diversity in several tasks like summarization (Kumari et al., 2024, Kothawade et al., 2022) and subset selection (Jain et al., 2023, Lin and Bilmes, 2011) to name a few. What is the justification of introducing only $F_{M}^{DiSF}(U)$ (Equation 7 in the paper) rather than any known submodular functions ? The The authors should compare the performance of DiSF when the underlying function is chosen to be some/all of the popular submodular functions proven to be effective for diversity selection.

4. Although the experimental results in Table 2 show a strong average performance across datasets, when the budget increases from 10B to 50B (5x increase in number of available training examples) the performance gain (overall) is quite small. Did the authors perform any additional experiments or theoretical analysis to justify this result ?

Some minor questions/ suggestions -
1. Section 2.3 can be moved higher up in the paper or a brief explanation on the cause and effects of dimensionality collapse should be highlighted in Section 1 for better readability.

2. Phrases like "Diversified Submodular File Selection" (line 19, 88, 183-184 ...), "decorrelated text files" (line 20, 89, 184 ...) has been repeated several times in the paper which should be reduced to improve readability.

3. Do the authors plan on releasing the code for reproducing the results from the paper in the near future ?

4.  In Equation 8 the $\text{argmax}$ is applied over $\mathbb{D} \setminus U$ which seems to be redundant. To the best of my knowledge, during the course of the greedy optimization process in Algorithm 1 of the paper, DiSF selects the sample with the highest information gain which inherently excludes duplicate samples which are already selected in U.

**Questions:**

Please refer to the weaknesses for detailed questions.

---

> ### Author Response · Authors · 2024-11-20
> **Rebuttal to H11o (1)**
>
> **Thanks for the constructive feedback provided by the Reviewer H11o. We sincerely appreciate the time and effort you dedicated to evaluating our work. Below, we provide detailed responses to the weaknesses and questions, which we denote as W and Q. Besides, we also added the following discussions into revised submission.**
> >**W1**: Although the subset selection mechanism in DiSF depends on submodular maximization of F but the proof of submodularity of F has not been provided. ..empirical proof of monotonicity is not deemed sufficient to prove a function to be submodular.
>
> **Reply to W1:** We thank for the constructive comments for improving our submission. In the original manuscript, we intend to provide approximation analysis when our function showing diminishing gain on set of feature (original description: 'our proxy function aims to capture the diversity in a given set, making it well-suited for submodular optimization'), instead of claiming it is a strictly submodular function on all sets. To enhance clarity and practicality, we have improved the descriptions in the revised submission, emphasizing that our function may not exhibit the submodular property across all file sets. Additionally, we employ more general $\gamma$-weakly submodular optimization properties to **empirically verify** the approximation to the optimal solution **during the execution of our algorithm, specifically on SlimPajama.** We provide more details in the following.
>
> 1) Revisions are:
> * We revised section 3 for better clarity. Specifically, we divide original section 3 into method and analysis, and move all contents related to submodular optimization to analysis. Besides, for a more concrete and practical verification to estimate its approximation to optimal solution during execution, we step further from submodular to $\gamma$-weakly submodular optimization (See details in the next item).
> * To avoid misunderstanding, we changed some section names from 'submodular selection' to 'diversified selection'.
> * To avoid misunderstanding, we changed the descriptions from 'model it as a submodular optimization problem' in title, abstract, and introduction to 'solve it with classical greedy algorithm and analyze the approximation from the view of $\gamma$-weakly submodular optimization'.
> 2) Non-negative monotone $\gamma$-weakly submodular optimization is more general for set functions, which provides a guarantee of $(1 − e^{-\gamma})$ approximation ($\gamma \in$(0,1]) [1] to the optimal solution with classical greedy algorithm. Submodular ratio $\gamma$ is defined as $\gamma = \min_{A \subseteq B \subseteq \Omega, e \notin B} \frac{f(A \cup \{e\}) - f(A)}{f(B \cup \{e\}) - f(B)},$ where $A\subset B\subset \Omega$.**When $\gamma=1$, the function is submodular over the given set. When $0<\gamma< 1$, the function exhibits weak submodularity over the given set. However, when $\gamma\leq 0$, the function is no longer suitable for analysis using this property.** In the revision, we estimate $\gamma$ during selection on SlimPajama in batch files to empirically verify the level of approximation. We estimate the submodular ratio $\gamma$ in 1000 selection batches. In each batch $\Omega=b_i$, we randomly sample 1000 cases of $A$, $B$ and $e$ from $b_i$, recording the smallest $\gamma$. **Although submodular ratio often shows very pessimistic bound in applications [2], as shown in the following Table T1, most cases are within (0,1] and show a quite large value, with an average submodular ratio $\hat{\gamma}=0.478$.** As shown in the following Table T2, we also calculate the gains $\delta_i$ after adding $i$ times of 32 selected samples across 5 different cases. All cases demonstrate diminishing returns in the gains. As shown in the following Table T2, we also calculate the gains $\delta_i$ after adding $i$ times of 32 selected samples for 5 cases. All cases demonstrate the diminishing gains.
>
> **We hope this provide more concrete and practical analysis of our algorithm and can address the concerns of the reviewer.**
>
> **Table T1** Estimation of submodular ratio $\gamma$ in 1000 batch files.
> |Statistics|0.8-1|0.6-0.8|0.4-0.6|0.2-0.4|0-0.2|<0|
> |-|-|-|-|-|-|-|
> | Num of cases|174|377|211|93|128|17|
>
> **Table T2** Diminishing return property during execution of algorithm.
> |Method|$\delta_1$|$\delta_2$|$\delta_3$|$\delta_4$|$\delta_5$|$\delta_6$|$\delta_7$|$\delta_8$|$\delta_9$|$\delta_{10}$|
> |-|-|-|-|-|-|-|-|-|-|-|
> |Case 1|0.099|0.069|0.063 |0.057|0.045|0.037|0.029|0.027|0.022|0.017|
> |Case 2|0.097|0.063|0.055|0.049|0.038|0.029|0.027|0.019|0.016|0.011|
> |Case 3|0.098|0.067|0.054|0.038|0.028|0.025|0.017|0.015|0.011|0.009|
> |Case 4| 0.097 | 0.077 |0.071|0.060|0.037 |0.024|0.023|0.017|0.011|0.008|
> |Case 5| 0.100|0.082 |0.079|0.055|0.047|0.025|0.020|0.019|0.014|0.007|
>
> [1] Submodular meets Spectral: Greedy Algorithms for Subset Selection, Sparse Approximation and Dictionary Selection
>
> [2] Weakly Submodular Function Maximization Using Local Submodularity Ratio

---

> > ### Author Response · Authors · 2024-11-20
> > **Rebuttal to H11o (2)**
> >
> > >**W2**: Diversity driven subset selection for data-efficient LLM pretraining is not novel to the current paper. Prior work like INGENIOUS (Renduchintala et al., 2023) have demonstrated significant improvements in model performance while being data-efficient. The authors should compare against such methods to study the impact of dimensionality collapse and the diversity selection algorithm.
> >
> > **Reply to W2:** **Thank you for the constructive comments and the promising recommended work. We have included INGENIOUS in our baselines for the revised submission.** INGENIOUS utilizes Facility Location and similarity kernels to select highly representative subset, which helps enhance the diversity of selected data. As shown in following Tables T2 and T3, we include INGENIOUS as a baseline and track its dominance score. Notably, we observed that INGENIOUS, when using a warmed-up model for feature extraction (INGENIOUS-W), does not perform well with our selected feature extractor, Contriever (INGENIOUS-C). From Table T2 and T3, INGENIOUS-C demonstrates competitive performance and dominance score among the baselines. However, our DiSF outperforms all methods, including INGENIOUS-C, by achieving both the best pre-trained performance and a significantly lower dominance score. **This is because INGENIOUS does not directly optimize the uniformity of feature dimensions as DiSF does, resulting in a trade-off between strict submodularity and the specific goal of optimizing dimension uniformity.**
> >
> > **Table T3** Comparison on dominance score $\frac{\sum_{i=1}^k\lambda_i}{\sum_{j=1}^d\lambda_j}$, where $\lambda_i$ is the $i$-th large eigenvalue of the covariance matrix, and $d$ is the dimension of the feature space.
> >
> > | | DSIR | QuRating-A | QuRating-W | D4 | INGENIOUS-W| INGENIOUS-C | DiSF (Ours) |
> > | - | - | - | - | - | - | - | - |
> > | k=20 |  0.319 | 0.192 | 0.279 | 0.240 | 0.290 | 0.179 |  0.129  |
> > | k=40 | 0.428 | 0.299 | 0.396 | 0.330 | 0.357 | 0.267 |  0.231  |
> > | k=60 |0.516 | 0.391 | 0.484 | 0.404 | 0.408 | 0.374 |  0.303  |
> > | k=80 |  0.589 | 0.470 | 0.557 | 0.468 | 0.496 | 0.432 |  0.371  |
> > | k=100 |  0.651 | 0.539 | 0.619 | 0.525 | 0.549 | 0.497 |  0.433  |
> >
> > **Table T4** Comparison on common sense ability with 50B training budget. More results is provided in the revised submission.
> > | Method     | Random | DSIR | QuRating-A | QuRating-W | Doremi | D4 | INGENIOUS-W| INGENIOUS-C | DiSF (Ours) |
> > | - | - | - | - | - | - | - | - | - | - |
> > | TinyLlama 120M     | 38.9|37.8|40.0|38.9|39.7|39.7|39.0| 39.6  | **40.6**|
> > | TinyLlama 560M     | 41.6|40.7|42.6|41.2|42.7|42.2|41.5| 42.3  |**43.2**|
> > | TinyLlama 1.1B     | 42.9|41.8|44.8|43.4|44.5|44.2|43.2| 44.1 |**45.2**|
> >
> > >**W3**: Submodular Functions like Log-Determinant, Facility-Location etc. have been shown to model diversity in several tasks like summarization (Kumari et al., 2024, Kothawade et al., 2022) and subset selection (Jain et al., 2023, Lin and Bilmes, 2011) to name a few. What is the justification of introducing only (Equation 7 in the paper) rather than any known submodular functions?
> >
> > **Thanks for the valuable advice to improve our submission. We have incorporated more submodular functions and all the mentioned works into the revised submission to provide a more detailed analysis of submodular functions, submodular optimization, and research works.** Furthermore, in response to W2, we include a recent method INGENIOUS based on Facility-Location into baseline. For Log-Determinant, we designed a variant defined as $F_{LD}$=LogDet(I+C) (DiSF-LD), where I is the identity matrix and C is covariance matrix defined in our original submission. We compare this variant with our original proxy function in following Table T5 on dominance score and Table T6 on pre-training performance for TinyLlama-120M, using a 1.5% selection ratio and a 50B training budget. Results demonstrate that DiSF-LD can help mitigate dimensional collapse and improve the performance of pre-trained LLMs. However, it does not achieve the same level of performance or dominance score as our original DiSF. Similar to the comparison with INGENIOUS, this is because DiSF-LD cannot directly optimize the uniformity of feature dimensions.
> >
> >  **Table T5** Dominance score compared to DiSF-LD.
> > | Setting| DSIR|D4|QuRating-W|QuRating-A|DiSF-LD|DiSF|
> > | - | - | - | - | - | - | - |
> > | k=20 |  0.319 | 0.240 | 0.279 | 0.192 | 0.220 |0.129 |
> > | k=40 | 0.428 | 0.330 | 0.396 | 0.299 | 0.325 |0.231 |
> > | k=60 |  0.516 | 0.404 | 0.484 | 0.391 | 0.425 |0.303 |
> > | k=80 | 0.589 | 0.468 | 0.557 | 0.470 | 0.476 |0.371 |
> > | k=100 |  0.651 | 0.525 | 0.619 | 0.539 | 0.534|0.433 |
> >
> > **Table T6** Pre-trained performance compared to DiSF-LD.
> > | Setting| Random |DSIR|D4|QuRating-W|QuRating-A|DiSF-LD|DiSF|
> > | - | - | - | - | - | - | - | - |
> > | TinyLlama120M| 38.9|37.8| 39.7 |38.9|40.0|39.5|**40.6**|

---

> > > ### Author Response · Authors · 2024-11-20
> > > **Rebuttal to H11o (3)**
> > >
> > > >**W4**: Although the experimental results in Table 2 show a strong average performance across datasets, when the budget increases from 10B to 50B (5x increase in number of available training examples) the performance gain (overall) is quite small. Did the authors perform any additional experiments or theoretical analysis to justify this result ?
> > >
> > > **Reply to W4:**
> > > We provide additional analysis about it from the following three points:
> > > 1) **It is not easy to improve the performance of pre-trained LLM on these popular tasks.** According to the **official TinyLlama report**, a 1.1B model pre-trained on 100B to 500B tokens across all data achieved an average performance improvement from 46.11 to 48.28 on seven common-sense tasks—a 2.1% gain. In our experiments, pre-training the same model with 10B to 50B tokens on 10B data, our method achieves a 3% improvement (from 42.2 to 45.2), while DSIR achieves a 1.7% improvement.
> > > 2) **Diversified samples achieve larger improvements with an increasing training budget.** The increased budget in Table 2 refers only to the training budget, while the selection budget remains fixed at 1.5% (approximately 10B). Methods such as DSIR, and QuRating-W tend to select similar files, increasing the risk of overfitting and limiting the model's ability to learn new information. In contrast, approaches like D4 and ours promote diversity, enabling the model to benefit from more varied data. Notably, our method achieves a 2.2% improvement, compared to a 1.4% gain with DSIR, underscoring the importance to enhance diversity.
> > > 3) **LLM with larger scale achieve larger improvement when rasing training budget.** Another factor may be the model's capacity. As shown in Table 2 of submission, when the model size is increased from 120M to 1.1B, the performance gain of our method rises from 1.7% to 3%, while the gain for DSIR increases from 0.9% to 1.7%. This suggests that larger models benefit more from pre-training on more tokens.
> > >
> > > >**W5**: Section 2.3 can be moved higher up in the paper or a brief explanation on the cause and effects of dimensionality collapse should be highlighted in Section 1 for better readability.
> > >
> > > **Reply to W5:**
> > > **Thanks for the valuable suggestion! We have included a brief explanation of the causes and effects of dimensional collapse in Section 1 of the revised submission.**
> > >
> > > >**W6**: Phrases like "Diversified Submodular File Selection" (line 19, 88, 183-184 ...), "decorrelated text files" (line 20, 89, 184 ...) has been repeated several times in the paper which should be reduced to improve readability.
> > >
> > > **Reply to W6:**
> > > Thanks for the advice! We have **removed repeated descriptions** and revised line 88, 89, 183, and 184 in the revised submission for better readability.
> > >
> > > >**W7**: Do the authors plan on releasing the code for reproducing the results from the paper in the near future ?
> > >
> > > **Reply to W7:**
> > > One of the contributions highlighted in our paper is 'a benchmark on TinyLlama architectures and the SlimPajama text corpus using 8 GPUs with 24GB memory' to facilitate common academic research on file selection in LLM pre-training. If accepted, **we will release the code, including pre-training TinyLlama, data preprocessing, and file selection based on our method and other baselines to ensure reproducibility.**
> > >
> > > >**W8**: In Equation 8 the argmax is applied over $\Omega / U$  which seems to be redundant. To the best of my knowledge, during the course of the greedy optimization process in Algorithm 1 of the paper, DiSF selects the sample with the highest information gain which inherently excludes duplicate samples which are already selected in U.
> > >
> > > **Reply to W8:**
> > > Although the selected results remain the same, applying argmax on $\Omega / U$ instead of $\Omega$ can **reduce the computational cost** of the proxy function for files that have already been selected **in the execution of the code**.

---

> > > > ### Author Response · Authors · 2024-11-23
> > > > **Invitation to rolling discussion for the possible remaining concerns**
> > > >
> > > > Dear Reviewer H11o,
> > > >
> > > > **We sincerely appreciate the effort and time you have devoted to providing constructive reviews. We have now provided more clarifications, explanations and experimental results to address your concerns. Here is a summary for you:**
> > > >
> > > > * We have clarified and refined several descriptions for improved readability.
> > > > * We further utilize the more general weakly submodular optimization property to conduct a practical analysis of our selection algorithm's execution.
> > > > * We incorporate another baseline, INGENIOUS, and compare our DiSF with two submodular functions.
> > > > * We include additional recent studies, introducing more submodular functions and related research.
> > > > * We provide a detailed analysis of performance gains when increasing the training budget.
> > > > * We promise to make experiments and method open source.
> > > >
> > > > Please let us know if anything is unclear. **We truly appreciate this opportunity to improve our work and shall be most grateful for any feedback you could give to us.**

---

> > > > > ### Comment · Reviewer_H11o · 2024-11-25
> > > > > **Thanks to Authors and Response to Rebuttals**
> > > > >
> > > > > I would like to thank the authors for putting in a detailed rebuttal and it is commendable that the authors have put in the extra effort to include additional methods and detailed empirical evidence supporting their proposed methodology. Purely on the basis of empirical evidence, I have increased my rating for this paper.
> > > > >
> > > > > But, my concerns on the theoretical guarantee, particularly **regarding $F^{DiSF}_M(U)$ to be $\gamma$-weakly submodular (discussed in response to *W1*)** continue to hold. Although the authors modify their claim in the paper from $F^{DiSF}_M(U)$ being *submodular* to being *$\gamma$-weakly submodular* and show empirical evidence, it is not sufficient theoretical guarantee on the application of DiSF in file selection tasks. The authors must provide a formal proof of $F^{DiSF}_M(U)$ being $\gamma$-weakly submodular for any given subset $A \subseteq U$, selected from the ground set $U$. This can be shown by proving that for $F^{DiSF}_M(U)$ the inequality $f(A \cup x) - f(A) \geq \gamma(f(B \cup x) - f(B))$ for a new element $x$ and $A \subseteq B \subseteq U$. Following this, the authors should show that all properties of $\gamma$-weakly submodular functions hold (Chen et al., 2018) for $F^{DiSF}_M(U)$, particularly guaranteeing the modelling of diversity when optimized using greedy maximization (a key milestone for DiSF).

---

> ### Author Response · Authors · 2024-11-26
> **Response to further question.**
>
> **We are grateful for the endorsement of the reviewer and the increased score.** We here to provide a more concrete theoretical guarantee on the submodular ratio.
>
> We first recall our function, which is based on F-norm of the covariance matrix. The gain on A, and B with another sample e, can be formulated as following:
>
> $\Delta_A = e^{-\frac{1}{|A| + 1} \sqrt{\sum_{i=1}^d \sum_{j=1}^d \left( \sum_{x \in A} x_i x_j + e_i e_j \right)^2}} - e^{-\frac{1}{|A|} \sqrt{\sum_{i=1}^d \sum_{j=1}^d \left( \sum_{x \in A} x_i x_j \right)^2}}$,
> $\Delta_B = e^{-\frac{1}{|B| + 1} \sqrt{\sum_{i=1}^d \sum_{j=1}^d \left( \sum_{x \in B} x_i x_j + e_i e_j \right)^2}} - e^{-\frac{1}{|B|} \sqrt{\sum_{i=1}^d \sum_{j=1}^d \left( \sum_{x \in B} x_i x_j \right)^2}}$,
>
> where $A\subset B \subset \Omega$, $e\in \Omega$, and $e \notin B$. We reformulate them as
>
> $\Delta_A = e^{-\frac{1}{|A|} \sqrt{\sum_{i}^d \sum_{j}^d (\sum_{x\in A}x_i x_j)^2}} (e^{\frac{1}{|A|} \sqrt{\sum_{i}^d \sum_{j}^d (\sum_{x\in A}x_i x_j)^2}-\frac{1}{|A+1|} \sqrt{\sum_{i}^d \sum_{j}^d (\sum_{x\in A}x_i x_j+e_ie_j)^2}} - 1)$,
> $\Delta_B = e^{-\frac{1}{|B|} \sqrt{\sum_{i}^d \sum_{j}^d (\sum_{x\in B}x_i x_j)^2}} (e^{\frac{1}{|B|} \sqrt{\sum_{i}^d \sum_{j}^d (\sum_{x\in B}x_i x_j)^2}-\frac{1}{|B+1|} \sqrt{\sum_{i}^d \sum_{j}^d (\sum_{x\in B}x_i x_j+e_ie_j)^2}} - 1)$
>
> Therefore, we have:
>
> $\frac{\Delta_A}{\Delta_B}=e^{\frac{1}{|B|} \sqrt{\sum_{i}^d \sum_{j}^d (\sum_{x\in B}x_i x_j)^2}-\frac{1}{|A|} \sqrt{\sum_{i}^d \sum_{j}^d (\sum_{x\in A}x_i x_j)^2}}*\frac{e^{\frac{1}{|A|} \sqrt{\sum_{i}^d \sum_{j}^d (\sum_{x\in A}x_i x_j)^2}-\frac{1}{|A+1|} \sqrt{\sum_{i}^d \sum_{j}^d (\sum_{x\in A}x_i x_j+e_ie_j)^2}} - 1}{e^{\frac{1}{|B|} \sqrt{\sum_{i}^d \sum_{j}^d (\sum_{x\in B}x_i x_j)^2}-\frac{1}{|B+1|} \sqrt{\sum_{i}^d \sum_{j}^d (\sum_{x\in B}x_i x_j+e_ie_j)^2}} - 1}$
>
> We then define:
>
> $\Delta(e|A)=\frac{1}{|A+1|} \sqrt{\sum_{i}^d \sum_{j}^d (\sum_{x\in A}x_i x_j+e_ie_j)^2}-\frac{1}{|A|} \sqrt{\sum_{i}^d \sum_{j}^d (\sum_{x\in A}x_i x_j)^2}$
>
> $\Delta(e|B)=\frac{1}{|B+1|} \sqrt{\sum_{i}^d \sum_{j}^d (\sum_{x\in B}x_i x_j+e_ie_j)^2-\frac{1}{|B|} \sqrt{\sum_{i}^d \sum_{j}^d (\sum_{x\in B}x_i x_j)^2}}$
>
> Therefore, the original equation will be:
> $\frac{\Delta_A}{\Delta_B}=e^{\frac{1}{|B|} \sqrt{\sum_{i}^d \sum_{j}^d (\sum_{x\in B}x_i x_j)^2}-\frac{1}{|A|} \sqrt{\sum_{i}^d \sum_{j}^d (\sum_{x\in A}x_i x_j)^2}} *\frac{e^{-\Delta(e|A)}-1}{e^{-\Delta(e|B)}-1}$.
>
> To provide a theoretical guarantee, we aim to establish a lower bound for the submodular ratio under any given set where the gain is positive, as most empirical verifications suggest. Since the original formulation is challenging to analyze directly, we introduce two bounds on the function applied to the text files to derive a more concrete lower bound, as follows:
>
> **Assumption 1)** Assume $|\Delta(e|A)-\Delta(e|B)|\leq \epsilon$, which means the gain difference between any two sets ($A\subset B \subset \Omega$) are bounded.
>
> **Assumption 2)** Assume average utility is $\mu$, which means $\forall U \in \Omega$, we have: $\frac{1}{|U|} \sqrt{\sum_{i}^d \sum_{j}^d (\sum_{x\in U}x_i x_j)^2}\leq \mu.$
>
> With Assumption 1, we have:
> $\frac{\Delta_A}{\Delta_B}\geq e^{\frac{1}{|B|} \sqrt{\sum_{i}^d \sum_{j}^d (\sum_{x\in B}x_i x_j)^2}-\frac{1}{|A|} \sqrt{\sum_{i}^d \sum_{j}^d (\sum_{x\in A}x_i x_j)^2}} *\frac{e^{-\Delta(e|B)-\epsilon}-1}{e^{-\Delta(e|B)}-1}$.
>
>
> With Assumption 2, we have:$\frac{1}{|B|} \sqrt{\sum_{i}^d \sum_{j}^d (\sum_{x\in B}x_i x_j)^2}-\frac{1}{|A|} \sqrt{\sum_{i}^d \sum_{j}^d (\sum_{x\in A}x_i x_j)^2}\geq -2\mu$, which means $\frac{\Delta_A}{\Delta_B}\geq e^{-2\mu} *\frac{e^{-\Delta(e|B)-\epsilon}-1}{e^{-\Delta(e|B)}-1}.$We reformulate this as
>
> $\frac{\Delta_A}{\Delta_B}\geq e^{-2\mu} *(e^{-\epsilon}+\frac{e^{-\epsilon}-1}{e^{-\Delta(e|B)}-1}).$
>
> From Assumption 2, we have $\frac{1}{|B|} \sqrt{\sum_{i}^d \sum_{j}^d (\sum_{x\in B}x_i x_j)^2}\leq \mu$. Therefore $e^{-\Delta(e|B)}\leq e^{2\mu}$, and we have:
> $\frac{\Delta_A}{\Delta_B}\geq e^{-2\mu} *(e^{-\epsilon}+\frac{e^{-\epsilon}-1}{e^{2\mu}-1}).$
>
> **Finally, we have a lower bound as $e^{-2\mu}\frac{e^{2\mu-\epsilon}-1}{e^{2\mu}-1}$, which means under defined assumptions, our function is at least $e^{-2\mu}\frac{e^{2\mu-\epsilon}-1}{e^{2\mu}-1}$-weakly submodular function. It can be seen that, the theoretical results are influenced by the gain difference and the average utility. Larger difference or utility indicate a smaller lower bound.**

---

> > ### Comment · Reviewer_H11o · 2024-11-27
> > **Thanks to Authors**
> >
> > I would like to commend the response of the authors given the short discussion window. The new response does addresses my concerns regarding the theoretical guarantees behind the selection function of DiSF. Based on the newly provided evidence, I am increasing my score further towards acceptance.

---

> > > ### Author Response · Authors · 2024-11-27
> > > **Thanks for the reply and the increased score !**
> > >
> > > Thank you for your response and the increased score! We have gained valuable insights from our discussion with you. We truly appreciate your constructive suggestions and will continue to refine our paper based on your feedback.

---

### Official Review · Reviewer_2LFP · 2024-11-03

**Soundness:** 3
**Presentation:** 3
**Contribution:** 3
**Rating:** 8
**Confidence:** 3

**Summary:**

This paper introduces DiSF, a method aimed at selecting diverse training data to enhance the effectiveness of pre-training large language models (LLMs). The authors begin by analyzing recent data selection methods and identify a critical issue: the selected data often suffer from dimensional collapse in the feature space. This collapse leads to improved performance on domain-specific tasks but causes a significant degradation in general performance due to a lack of diversity.

To address this problem, the authors propose improving the diversity of the data selection process by formulating it as a submodular optimization problem. Specifically, their goal is to prevent dimensional collapse by ensuring more uniform eigenvalues in the covariance matrix of the selected samples. They employ a classical greedy algorithm for submodular optimization, combined with mini-batch sampling, to make the approach tractable.

Due to hardware limitations, the authors tested their approach on models with sizes of 120M, 560M, and 1.1B parameters. Despite the smaller scale, the DiSF method achieves superior performance over six baselines on nine standard benchmarks consistently, across different training budgets (10B and 50B tokens). The authors also provide extensive ablation studies to analyze the effects of data selection budget, different architectural backbones, feature extractors, and more.

**Strengths:**

+ Well-Motivated Research: The paper addresses a significant issue in data selection for LLMs—dimensional collapse due to lack of diversity—which has substantial implications for model performance.
+ Clear Presentation: The methodology is clearly explained, making the paper easy to read and understand.
+ Strong Experimental Results: The results are convincing, demonstrating that DiSF outperforms existing methods consistently across multiple benchmarks and model sizes.
+ Thoughtful Comparisons: Comparisons with six baseline methods are thorough, providing a solid context for the effectiveness of DiSF.
+ Comprehensive Ablation Studies: The extensive ablation studies offer valuable insights into the impact of various factors such as data selection budget, model architecture, and feature extractors.

**Weaknesses:**

+ Limited Scale of Experiments: The experiments are limited to models up to 1.1B parameters. While the results are promising, it would be beneficial to see how the method scales to larger models commonly used in current LLM research.
+ Scalability Concerns: There is uncertainty about whether the proposed method can maintain its efficiency and effectiveness when applied to larger datasets.

**Questions:**

The paper appears to lack a detailed analysis of the time and computational costs associated with the data selection process. How long does it take to complete the data selection using DiSF, and is it efficient enough to be practical for larger datasets?

---

> ### Author Response · Authors · 2024-11-20
> **Rebuttal to Reviewer 2LFP  (1)**
>
> **Thanks for the constructive feedback provided by the Reviewer 2LFP. We sincerely appreciate the time and effort you dedicated to evaluating our work. Below, we provide detailed responses to the weaknesses and questions, which we denote as W and Q. Besides, we also added the following discussions into revised submission.**
>
> > **W1 and W2**: 1) While the results are promising, it would be beneficial to see how the method scales to larger models commonly used in current LLM research. 2) Whether the proposed method can maintain its efficiency and effectiveness when applied to larger datasets.
>
> **Reply to W1 and W2:**
>
> To evaluate the effectiveness of our selection method on larger models and datasets, we **try our best to 1) pre-train Open-Llama 3B [1] on SlimPajama, and 2) TinyLlama-1B on both SlimPajama and StarcoderData [2].** StarcoderData is a dataset created for code generation, containing approximately 210M files (500GB of data). We assess the pre-trained models on common sense ability (7 tasks), mmlu, bbh and an additional programming task, code_x_glue [3], using harness evaluation. **Due to the limited rebuttal time, the cost of pre-training, and the effort involved in selecting data under other baselines, we present the most results we can at this stage. We hope this can address the reviewer’s concerns.** Comparison to QuRating in some settings are still ongoing, as it requires running its judger model (QuRater-1.3B) to evaluate all files, and we expect to complete it in the coming days.
> 1) As shown in the following Table T1, we pre-train open-llama 3B with 10B training budget and 1.5% selection budget with SlimPajama (on 4xA100 devices with about 4 days) . We compare our method with Random, DSIR, QuRating-A, and QuRating-W. DSIR and QuRating-W meet performance degradation due to dimensional collapse, while our method achieves the best performance on all tasks.
> 2) As shown in the following Table T2, we pre-train TinyLlama-1B on both SlimPajama and StarcoderData with 1.5% selection budget and 10B training budget, compared to random and DSIR. Performance of DSIR and QuRating-W also meet performance degradation in code generation ability. We analyze that DSIR tends to ignore code files in StarcoderData because its selection criterion is based on Wikipedia, leading to dimensional collapse. Besides, our DiSF can mitigate this collapse and achives the best performance on all tasks.
>
> Notably, in Table 4 of our original submission (Table 5 in the revised submission), we also evaluate our method on Pythia-1B and OPT-1.3B in terms of average commonsense reasoning performance, which might also helps illustrate the scalability of our method on current models in the LLM community.
>
>
> **Table T1**: Comparison on openllama 3B with 1.5% selecting ratio and 10 B pre-training budget on slimpajama.
> | Setting       | Random | DSIR | QuRating-W | QuRating-A | DiSF (Ours) |
> | ------------- | ---------------- | ---- | ---------- | ---------- | ----------- |
> | arc_challenge | 20.1   | 22.3  | **24.2** | 23.3 | 22.1  |
> | arc_easy      | 47.1   | 46.6  | 49.2 | **52.1** | 49.1  |
> | boolq         | 60.6   | 59.5  | 60.1 | 61.0  | **61.8**  |
> | hellaswag     | 33.9   | 29.7  | 30.7| 33.2 | **34.4**  |
> | openbookqa    | 19.8   | 19.6  | 20.0 | 20.4| **20.6**  |
> | piqa          | 67.7   | 65.5  | 62.9 | 63.0 | **69.9**  |
> | winogrande    | 52.2   | 49.8  | 50.9 | 52.3  | **52.8** |
> | **Avg**       | 43.1   | 41.9  | 42.6   | 43.6| **44.4** |
>
>
> **Table T2**: Comparison on TinyLlama-1B with 1.5% selecting ratio and 10 B pre-training budget on both slimpajama and starcoderdata.
> | Setting     | Full-Data/Random | DSIR | DiSF (Ours) |
> | ----------- | ---------------- | ---- | ----------- |
> |common sense  |  40.3  |   40.0   |  **41.4**  |
> |bbh |  12.6  |    12.5  |  **13.3**  |
> |mmlu |  23.0  |   22.1   |   **23.7**  |
> |code_x_glue |0.80|0.59|**0.86** |
>
> [1] Openllama: An open reproduction of llama
>
> [2] Starcoder: may the source be with you!
>
> [3] CodeXGLUE: A Machine Learning Benchmark Dataset for Code Understanding and Generation

---

> > ### Author Response · Authors · 2024-11-20
> > **Rebuttal to Reviewer 2LFP (2)**
> >
> > > **Q1**: The paper appears to lack a detailed analysis of the time and computational costs associated with the data selection process. How long does it take to complete the data selection using DiSF, and is it efficient enough to be practical for larger datasets?
> >
> > **Reply to Q1:**
> > 1) We report the **computational time** for selection in Figure 7 of the original submission as the selection ratio b changes. In our experimental settings, DiSF completed the selection of 590M training files (nearly **1TB of data**) in approximately **26 hours using a single 4090 GPU and 48 CPUs**, with further acceleration possible by adding more CPUs. In pre-training tasks on large datasets and models with carefully tuning parameters, such selection level is acceptable.
> > 2) For **time complexity, we achieve at most $O(|S|^2bd^2)$. All terms in the time complexity are at most quadratic and independent of the overall dataset size, which we consider acceptable for applying to larger datasets.** In the following, we provide a detaild analysis. We divide it into two parts: 1) Computational complexity shown in Algorithm 1 of the submission. In each batch, we initialize
> >  $\mathrm{U}$ with a randomly selected sample and remove it from the batch. Then, we iteratively apply $(\lfloor\frac{b|\mathcal{S}|}{|\mathbb{D}|}\rfloor-1)$ times the Argmax command on the batch of data with our proxy fuction. Denote the computational cost of our proxy function with k text samples as $F_{|U|=k}(U)=O(F_k)$, the computation cost will be:$$O(1+...+\frac{|\mathbb{D}|}{b})\sum_{k=1}^{\frac{b|\mathcal{S}|}{|\mathbb{D}|}}(b-k)(F_{k+1}) \leq O(\frac{|\mathbb{D}|^2}{b^2})\frac{b|\mathcal{S}|}{|\mathbb{D}|}b F_{k+1} =O(|\mathbb{D}|\mathcal{S}F_{\frac{b|\mathcal{S}|}{|\mathbb{D}|}+1}),$$ where b is the batch scale, $|\mathbb{D}|$ is the total data scale, $|\mathcal{S}|$ is the selection budget. 2) The complexity of proxy function $O(F_k)$. Given text features $z$ and their feature dimension d, Frobenius norm and $z\cdot z^T$ are both $O(d^2)$. Since our proxy function calculates k times the $z\cdot z^T$, therefore $O(F_k)=kd^2$. Finally, the complexity of our DiSF will be: $$O(DiSF)\leq O(|\mathbb{D}|\mathcal{S}F_{\frac{b|\mathcal{S}|}{|\mathbb{D}|}+1}) =O(|S|^2bd^2)$$
> >
> > Notably, as reported in QuRating, annotating the data using the GPT API costs 520 NVIDIA H100 hours with additional ranking procedures of 3 hours. For DSIR, it takes more than 2 days using 48 CPUs in our platform. For DOREMI, training a 120M proxy model to provide weights for domains takes us approximately one week. In contrast, our method utilizes a public feature extractor and selects samples in about 26 hours using one GPU and 48 CPUs. **Combining these facts and our complexity analysis, we believe our method is practical among these methods for larger datasets.**

---

> > > ### Author Response · Authors · 2024-11-23
> > > **Invitation to rolling discussion for the possible remaining concerns**
> > >
> > > Dear Reviewer 2LFP,
> > >
> > > Thank you again for your time and efforts in reviewing our submission, as well as your positive evaluation of our submission! **We have now provided more clarifications, explanations, experiments, and discussions to address your concerns and followed your advice to improve our paper, including:**
> > >
> > > * We verify our method on a larger model, Open-Llama-3B.
> > > * We validate our approach on a larger dataset, combining SlimPajama and Starcoderdata.
> > > * We provide a detailed analysis of the time and computational costs involved in the data selection process.
> > >
> > > Would you mind checking our responses and confirming if they have addressed your concerns? **We truly appreciate this opportunity to improve our work and shall be most grateful for any feedback you could give to us.**

---

> > > ### Comment · Reviewer_2LFP · 2024-11-26
> > > **Reply to the author**
> > >
> > > Thank you for your answer. I would like to see this paper at ICLR, and I have raised my rating.

---

> > > > ### Author Response · Authors · 2024-11-27
> > > > **Thanks for the reply and the increased score !**
> > > >
> > > > We are grateful for the endorsement of the reviewer and the increased score! We will keep improving our paper, following your valuable suggestions.

---

### Meta-Review · Area_Chair_7LGN · 2024-12-18

**Metareview:**

This paper presents DiverSified File selection algorithm (DiSF), a method for selecting high-quality pre-training data for large language models (LLMs) under computational constraints. Traditional file selection approaches focus on domain similarity, prioritizing data resembling high-quality sources (e.g., BookCorpus, Wikipedia). However, these methods suffer from a "diversity dilemma," where feature space collapse degrades generic performance despite improvements in domain-specific tasks. DiSF addresses this issue by selecting decorrelated text files to maintain diversity in the feature space. Using a greedy algorithm, it optimizes for uniform eigenvalues in the feature covariance matrix, treating the problem as weakly submodular optimization. The extensive experiments showcase that the approach seems to work well on several tasks resulting in a 1.5x training efficiency and a 5x data efficiency.

All the reviewers agree that this paper should be accepted at ICLR, and I agree with this assessment. I would like to encourage the authors to kindly take all the feedback into consideration including additional experiments and baselines to add to strengthen the paper.

**Additional Comments On Reviewer Discussion:**

All the reviewers agree that this paper should be accepted at ICLR, and I agree with this assessment. I would like to encourage the authors to kindly take all the feedback into consideration including additional experiments and baselines to add to strengthen the paper.

---

### Decision · Program_Chairs · 2025-01-22

Accept (Oral)